# BOOSTING ADVERSARIAL ROBUSTNESS AND GENERALIZATION WITH DICTIONARY STRUCTURE

## ABSTRACT

This work investigates a novel approach to boost adversarial robustness and generalization by incorporating structural prior into the design of deep learning models. Specifically, our study surprisingly reveals that existing dictionary learning-inspired convolutional neural networks (CNNs) provide a false sense of security against adversarial attacks. To address this, we propose Elastic Dictionary Learning Networks (EDLNets), a novel ResNet architecture that significantly enhances adversarial robustness and generalization. Extensive and reliable experiments demonstrate consistent and significant performance improvement on open robustness leaderboards such as RobustBench, surpassing state-of-the-art baselines. To the best of our knowledge, this is the first work to discover and validate that dictionary structure can reliably enhance deep learning robustness under strong adaptive attacks, unveiling a promising direction for future research.

## 1 INTRODUCTION

Adversarial robustness has become a central challenge in modern machine learning, particularly for deep neural networks deployed in high-stakes visual applications. Recent advances show that state-of-the-art defenses are predominantly built upon adversarial training (Madry, 2017; Zhang et al., 2019; Gowal et al., 2021) and various regularization strategies (Cisse et al., 2017; Zheng et al., 2016). In the visual domain, adversarial training combined with generative modeling (Wang et al., 2023; Gowal et al., 2021) has driven substantial progress and currently dominates the robustness leaderboard (Croce et al., 2020). However, these approaches increasingly rely on large amounts of synthetic data and ever-growing model capacity, suggesting a potential saturation of gains within this paradigm.

Despite their success, adversarially trained networks often improve robustness by memorizing adversarial perturbations (Madry, 2017), which makes them susceptible to the well-known problem of *robust overfitting* (Rice et al., 2020). Existing works attempt to alleviate robust overfitting through regularization (Andriushchenko & Flammarion, 2020; Qin et al., 2019; Sriramanan et al., 2020), data augmentation (DeVries, 2017; Zhang, 2017; Carmon et al., 2019; Zhai et al., 2019), or generative modeling techniques (Wang et al., 2023; Gowal et al., 2021). However, these methods operate within the same optimization and training framework, making further breakthroughs difficult without fundamentally new architectural principles.

Motivated by this observation, we explore an orthogonal direction grounded in the *dictionary structural prior* that has been widely studied in sparse coding and convolutional dictionary learning. Prior works (Papyan et al., 2017; Cazenavette et al., 2021; Mahdizadehaghdam et al., 2019; Li et al., 2022) suggest that natural signals can be represented as sparse linear combinations of learned atoms, enabling effective denoising of random corruptions and universal perturbations. Yet, this line of research has not been fully explored under strong, adaptive adversarial attacks, and its limitations in such settings remain under-investigated.

To address this gap, we revisit convolutional dictionary learning in the context of adversarial robustness and provide both empirical and theoretical analysis showing why existing dictionary-based architectures struggle under adaptive attacks. Building on these insights, we propose *Elastic Dictionary Learning (Elastic DL)*, a flexible framework that complements adversarial training and achieves improved robustness–generalization trade-offs. Our main contributions are summarized as follows:

- We revisit convolutional dictionary learning in deep learning, highlighting its failures under adaptive attacks, and we provide theoretical insights into these limitations.
- We first propose a robust dictionary learning approach via $\ell_1$-reconstruction and highlight its lower natural performance and the challenges in handling adaptive attacks. Furthermore, we introduce a novel Elastic Dictionary Learning (Elastic DL) framework to enable a better trade-off between natural and robust performance.
- We develop an efficient reweighted iterative shrinkage thresholding algorithm (RISTA) to approximate the non-smooth Elastic DL objective with theoretical convergence guarantees. The algorithm can be seamlessly integrated into deep learning models as a replacement for conventional convolutional layers to enhance all convolutional architectures.
- Extensive experiments demonstrate that our proposed Elastic DL framework can significantly improves adversarial robustness and generalization. Notably, our Elastic DL can achieve state-of-the-art performance, significantly outperforming the previous best defense on RobustBench (Croce et al., 2020) leaderboard across various budgets under $\ell_\infty$-norm and $\ell_2$-norm attacks.

## 2 RELATED WORKS

**Robust overfitting.** Overfitting in adversarially trained deep networks has been shown to significantly harm test robustness (Rice et al., 2020). To address the issue of severe robust overfitting, several efforts have been made from various perspectives. For instance, Dropout (Srivastava et al., 2014) is a widely used regularization method that randomly disables units and connections during training to mitigate overfitting. Regularization techniques (Andriushchenko & Flammarion, 2020; Qin et al., 2019; Sriramanan et al., 2020) have also proven effective in preventing overfitting by penalizing the complexity of model parameters. Data augmentation is another common approach for reducing overfitting in deep network training (Schmidt et al., 2018), with methods including Cutout (DeVries, 2017), Mixup (Zhang, 2017), semi-supervised learning techniques (Carmon et al., 2019; Zhai et al., 2019), and generative modeling (Wang et al., 2023; Gowal et al., 2021) being particularly notable. Additionally, early stopping (Rice et al., 2020) has demonstrated great effectiveness in achieving optimal robust performance during adversarial training. However, existing methods have yet to fully realize the potential of structural priors for improving adversarial robustness and generalization.

**Dictionary learning prior in deep learning.** Dictionary learning has been well-studied and widely applied in signal and image processing (Olshausen & Field, 1996; Wright et al., 2008; Wright & Ma, 2010; Zhao et al., 2011; Yang et al., 2011; Lu et al., 2013; Chen & Wu, 2013; Jiang et al., 2015; Yang et al., 2011), based on the assumption that an input signal can be represented by a few atoms from a dictionary. Building on this foundation, Papyan et al. (2017); Cazenavette et al. (2021); Mahdizadehaghdam et al. (2019); Li et al. (2022) successfully incorporated dictionary learning into deep learning to interpret or replace the "black-box" nature of neural networks. While these methods have demonstrated promising generalization and robustness against random noise and universal attacks (Li et al., 2022; Mahdizadehaghdam et al., 2019), their practical benefits for improving robustness under adaptive attacks are yet to be thoroughly investigated. We leave the related works about general adversarial attacks and defenses in the Appendix C due to the space limit.

## 3 REVISITING CONVOLUTIONAL DICTIONARY LEARNING IN DEEP LEARNING

**Notations.** Let the input signal be denoted as $\boldsymbol{\xi} \in \mathbb{R}^{H \times W}$ and the convolution kernel as $\boldsymbol{\alpha} \in \mathbb{R}^{k \times k}$, where $k = 2k_0 + 1$. The *convolution* and the *transposed convolution* of $\boldsymbol{\xi}$ and $\boldsymbol{\alpha}$ are defined as:

$$(\boldsymbol{\alpha} \star \boldsymbol{\xi})[i,j] = \sum_{p=-k_0}^{k_0} \sum_{q=-k_0}^{k_0} \boldsymbol{\xi}[i+p, j+q] \cdot \boldsymbol{\alpha}[p,q], \quad (\boldsymbol{\alpha} * \boldsymbol{\xi})[i,j] = \sum_{p=-k_0}^{k_0} \sum_{q=-k_0}^{k_0} \boldsymbol{\xi}[i-p, j-q] \cdot \boldsymbol{\alpha}[p,q].$$

Let the $C$-channel input signal be denoted as $\boldsymbol{x} = \{\boldsymbol{\xi}_1, ..., \boldsymbol{\xi}_C\} \in \mathbb{R}^{H \times W \times C}$, and $D$-channel the output signal as $\boldsymbol{z} = \{\boldsymbol{\eta}_1, ..., \boldsymbol{\eta}_D\} \in \mathbb{R}^{H \times W \times D}$. The convolution operator $\mathcal{A}(\cdot)$ and its adjoint transposed convolution operator $\mathcal{A}^*$ are associated with kernel $\mathbf{A}$ as:

$$\mathcal{A}(\boldsymbol{x}) = \sum_{c=1}^{C} (\boldsymbol{\alpha}_{1c} \star \boldsymbol{\xi}_c, ..., \boldsymbol{\alpha}_{Dc} \star \boldsymbol{\xi}_c), \quad \mathcal{A}^*(\boldsymbol{z}) = \sum_{d=1}^{D} (\boldsymbol{\alpha}_{d1} * \boldsymbol{\eta}_d, ..., \boldsymbol{\alpha}_{dC} * \boldsymbol{\eta}_d),$$

where the associated kernel $\mathbf{A} = \{\boldsymbol{\alpha}_{dc}\}_{d \in [D], c \in [C]} \in \mathbb{R}^{D \times C \times k \times k}$. Here, $H$, $W$, $C$, $D$, and $k$ represent the height, width, input dimension, output dimension, and kernel size, respectively.

## 3.1 Vanilla Dictionary Learning

To enhance the interpretability of black-box deep neural networks (DNNs), Papyan et al. (2017); Cazenavette et al. (2021); Mahdizadehaghdam et al. (2019); Li et al. (2022) introduce the structural prior of dictionary learning into the design of neural networks, assuming that the signal $\boldsymbol{x}$ can be represented by a linear superposition of several atoms $\{\boldsymbol{\alpha}_{dc}\}$ from a convolutional dictionary $\mathbf{A}$: $\boldsymbol{x} = \mathcal{A}^*(\boldsymbol{z}) \in \mathbb{R}^{H \times W \times C}$. Then a sparse code $\boldsymbol{z}$ is sought to extract few descriptors out of the collected dictionary for any given input $\boldsymbol{x}$:

$$\min_{\boldsymbol{z}} \|\boldsymbol{x} - \mathcal{A}^*(\boldsymbol{z})\|_2^2 + \lambda \|\boldsymbol{z}\|_1, \tag{1}$$

where $\lambda$ is the hyperparameter to balance the fidelity and sparsity terms. The underlying intuition is that the dictionary captures the intrinsic structure of clean data, enabling the model to filter out perturbations that are not consistent with this structure. Consequently, when inputs are corrupted by adversarial noise or outliers, the reconstruction process using the learned dictionary can act as a denoising mechanism, preserving essential features while suppressing irrelevant or malicious variations. Although several works (Cazenavette et al., 2021; Mahdizadehaghdam et al., 2019; Li et al., 2022) demonstrated promising robustness of this vanilla dictionary learning (Vanilla DL) defined in Eq. (1) against random corruptions and universal adversarial attacks, it remains unclear whether Vanilla DL can withstand stronger adaptive attacks.

## 3.2 Preliminary Study: Vanilla DL-based SDNets Is Not Truly Robust

To validate the robustness of Vanilla DL, we conduct a preliminary experiment on SDNet18 (Li et al., 2022), a variant of ResNet18 in which all convolutional layers are replaced with convolutional sparse coding (CSC) layers based on Vanilla DL in Eq. (1). We evaluate the SDNet18 (with fixed $\lambda$ and tuned $\lambda$) under both random impulse noise and adaptive PGD adversarial attack (Madry, 2017) with budget $\frac{8}{255}$.

As shown in Table 1, SDNet18 improves upon ResNet18 in terms of robustness against random noise, with more significant improvement achieved by tuning the sparsity weight $\lambda$. However, SDNet18 still experiences a sharp drop in performance under adaptive PGD attack, with accuracy approaching zero. The detailed results of the performance under various noise levels and $\lambda$ values are presented in Figure 13 in Appendix D.1.

Table 1: Preliminary study on SDNet18 (Li et al., 2022) under varying levels of random noise and PGD attack ($\epsilon = \frac{8}{255}$).

| Model \ Noise Level | L-1 | L-2 | L-3 | L-4 | L-5 | PGD |
|---|---|---|---|---|---|---|
| ResNet18 | 81.44 | 57.23 | 48.32 | 32.49 | 16.98 | **0.00** |
| SDNet18 ($\lambda = 0.1$) | 82.39 | 68.90 | 59.28 | 40.8 | 23.83 | **0.01** |
| SDNet18 (Tune $\lambda$) | 82.39 | 68.90 | 59.28 | 43.71 | 33.43 | **0.13** |

In fact, the $\ell_2$-reconstruction term of Vanilla DL in Eq. (1) imposes a quadratic penalty $\| \cdot \|_2^2$ on the residual $\boldsymbol{x} - \mathcal{A}^*(\boldsymbol{z})$, making it highly sensitive to outliers introduced by high-level noise and adaptive attacks. The experimental results reveal that existing Vanilla DL gives a *false sense of security* under random noise and can easily compromised by adaptive attack. Thus, there still remains a huge gap to achieve truly robust dictionary learning in deep learning.

## 4 Elastic Dictionary Learning

To overcome the aforementioned limitation brought by the Vanilla DL models, we first propose a robust dictionary learning (Robust DL) via $\ell_1$-reconstruction to mitigate the impact of outlying values in Section 4.1. Moreover, we conduct a comprehensive experiment to demonstrate the advantages of Robust DL and highlight its pitfalls in Section 4.2. Furthermore, to achieve a better inherent trade-off between natural and robust performance, we propose a novel elastic dictionary learning (Elastic DL) approach that enhances both natural performance and robustness in Section 4.3. The overview of our Elastic DL networks (EDLNets) can be found in Figure 1.

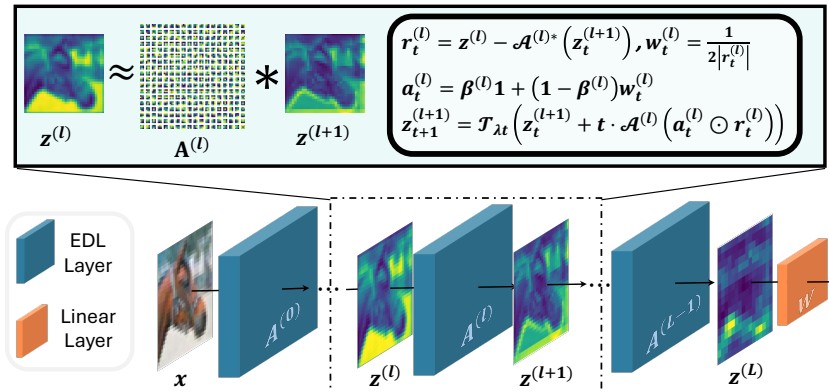

Figure 1: Overview of Elastic DL Networks (EDLNets). EDLNets are constructed by replacing the convolutional layers in backbones (e.g., ResNets) with EDL layers that are unrolled with the proposed efficient RISTA algorithm. Each EDL layer introduces a dictionary structural prior, assuming the input signal $z^{(l)}$ is encoded as a sparse code $z^{(l+1)}$ using a few atoms from diction $\mathbf{A}^{(l)}$.

## 4.1 ROBUST DICTIONARY LEARNING VIA $\ell_1$-RECONSTRUCTION

As observed in the previous section, $\ell_2$-fidelity assumes light-tailed noise and performs poorly as the noise becomes increasingly heavy-tailed. To address the sensitivity of $\ell_2$-fidelity in Vanilla DL, we first propose a robust dictionary learning approach (Robust DL) with $\ell_1$-reconstruction to effectively mitigate the impact of outliers:

$$\min_{z} \|x - \mathcal{A}^*(z)\|_1 + \lambda\|z\|_1. \tag{2}$$

Despite the sophisticated design of the model architecture, the $\ell_1$-norm terms in Eq.(2) introduce non-smoothness to the objective function, making it challenging to design an effective and efficient algorithm for approximating the solution. To address this, we first propose a *localized upper bound* as an alternative objective for the $\ell_1$-fidelity term $\|x - \mathcal{A}^*(z)\|_1$. Subsequently, we employ the iterative shrinkage-thresholding algorithm (ISTA) to solve the $\ell_1$-sparsity.

**Localized upper bound.** To address $\|x - \mathcal{A}^*(z)\|_1$ term, we first propose a convex upper bound $\mathcal{U}(z, z_*)$ as an alternative in the following Lemma 4.1.

**Lemma 4.1.** *Let $\mathcal{R}(z) := \|x - \mathcal{A}^*(z)\|_1$, and for any fixed point $z_*$, $\mathcal{U}(z, z_*)$ is defined as*

$$\mathcal{U}(z, z_*) = \|w^{1/2} \odot (x - \mathcal{A}^*(z))\|_2^2 + \mathcal{R}(z_*), \tag{3}$$

*where $w = \frac{1}{2|x - \mathcal{A}^*(z_*)|}$. Then, for any $z$, the following holds:*

$$(1)\ \mathcal{U}(z, z_*) \geq \mathcal{R}(z), \quad (2)\ \mathcal{U}(z_*, z_*) = \mathcal{R}(z_*).$$

*Proof.* Please refer to Appendix B.1. □

The statement (1) indicates that $\mathcal{U}(z, z_*)$ serves as an upper bound for $\mathcal{R}(z)$, while statement (2) demonstrates that $\mathcal{U}(z, z_*)$ equals $\mathcal{R}(z)$ at point $z_*$. With fixed $z_*$, the alternative objective $\mathcal{U}(z, z_*)$ in Eq. (3) is quadratic and can be efficiently optimized. Therefore, instead of minimizing the non-smooth $\mathcal{R}(z)$ directly, we can alternatively optimize the quadratic upper bound $\mathcal{U}(z, z_t)$ with gradient descent algorithm at iteration $t$.

**Reweighted ISTA (RISTA) algorithm.** According to Lemma 4.1, we can find an alternative objective for Eq. (2) at each step $t$:

$$z_{t+1} = \arg\min_{z} \|w_t^{1/2} \odot (x - \mathcal{A}^*(z))\|_2^2 + \lambda\|z\|_1, \tag{4}$$

where $w_t = \frac{1}{2|x - \mathcal{A}^*(z_t)|} \in \mathbb{R}^{H \times W \times C}$. Specifically, when $w_t = \mathbf{1}$, the problem reduces to the formulation in Eq. (1). Then, we can optimize the $\ell_1$-regularized problem in Eq. (4) instead of original Eq. (2) by our reweighted iterative shrinkage thresholding algorithm (RISTA):

$$z_{t+1} = \mathcal{T}_{\lambda t}(z_t + t \cdot \mathcal{A}(w_t \odot (x - \mathcal{A}^*(z_t)))), \tag{5}$$

where $\mathcal{T}_{\lambda t}(z) = \text{sign}(z)(|z - \lambda t|)_+$ represents the soft thresholding operator. The detailed derivation of Eq. (5) is provided in Appendix B.2. As a consequence of Lemma 4.1, we can conclude the iteration $\{z_t\}_{t=0}^T$ obtained by Eq. (5) fulfill the loss descent of $\mathcal{R}(z) + \|z\|_1$:

$$\mathcal{R}(z_{t+1}) + \|z_{t+1}\|_1 \leq \mathcal{U}(z_{t+1}, z_t) + \|z_{t+1}\|_1 \leq \mathcal{U}(z_t, z_t) + \|z_t\|_1 = \mathcal{R}(z_t) + \|z_t\|_1.$$

This implies convergence of Eq. (2) can be achieved by optimizing the localized upper bound Eq. (4).

### 4.2 Pitfalls in $\ell_1$-based Robust DL

To demonstrate the advantages of Robust DL over Vanilla DL, we evaluate the models under random noise and adaptive PGD attacks with attack budgets measured in $\ell_\infty$ and $\ell_2$ norms. From Table 2, we observe that $\ell_1$-based Robust DL has the following pitfalls:

- **Pitfall 1: Limited robustness.** In terms of robustness, Robust DL demonstrates a significant advantage over Vanilla DL under high-level random noise and adaptive adversarial attacks (PGD-$\ell_\infty$ and PGD-$\ell_2$) across various budget levels. However, both methods remain vulnerable to adver-

Table 2: Vanilla DL vs. Robust DL under random corruption (Impulse noise), PGD-$\ell_\infty$ and PGD-$\ell_2$ with various noise levels. Robust DL demonstrates significant improvement over Vanilla DL in robustness but sacrifices natural performance as a trade-off.

| RANDOM | NATURAL | L-1 | L-2 | L-3 | L-4 | L-5 |
|---|---|---|---|---|---|---|
| VANILLA DL | 93.38 | 84.95 | 75.83 | 67.22 | 44.01 | 24.91 |
| ROBUST DL | 83.25 | 77.71 | 71.69 | 64.9 | 51.02 | 37.78 |
| PGD-$\ell_\infty$ | NATURAL | 1/255 | 2/255 | 3/255 | 4/255 | 8/255 |
| VANILLA DL | 93.38 | 59.33 | 12.64 | 1.65 | 0.33 | 0.01 |
| ROBUST DL | 83.25 | 64.16 | 37.76 | 18.64 | 8.10 | 0.20 |
| PGD-$\ell_2$ | NATURAL | 0.1 | 0.2 | 0.3 | 0.4 | 0.6 |
| VANILLA DL | 93.38 | 63.61 | 27.86 | 9.78 | 3.31 | 0.10 |
| ROBUST DL | 83.25 | 69.56 | 50.17 | 32.58 | 20.25 | 2.79 |

sarially crafted perturbations, achieving nearly zero accuracy under adaptive attacks with imperceptible budgets (8/255 for PGD-$\ell_\infty$ and 0.6 for PGD-$\ell_2$).

- **Pitfall 2: Natural performance sacrifice.** Despite of certain improvement in robustness, Robust DL sacrifices natural performance by $10.13\%$. We conjecture that although $\ell_1$-based Robust DL effectively mitigates the impact of outlying values, it also misses important information due to the tradeoff between accuracy and robustness.

### 4.3 Elastic Dictionary Learning

From previous section, we can see that it is not trivial to design an optimal dictionary learning framework with either $\ell_2$ or $\ell_1$ reconstruction alone. To this end, we propose an elastic dictionary learning (Elastic DL) to achieve well-balanced trade-off between natural and robust performance:

$$\min_z \frac{\beta}{2}\|x - \mathcal{A}^*(z)\|_2^2 + \frac{1-\beta}{2}\|x - \mathcal{A}^*(z)\|_1 + \lambda\|z\|_1, \quad (6)$$

where $\beta$ is a layer-wise learnable parameter to adaptively balance the two fidelity terms. Similarly, we can generalize the RISTA algorithm from Robust DL to Elastic DL as in Appendix B.2. The RISTA algorithm for the Elastic DL layer is presented in Algorithm 1, and an overview of the entire EDLNet architecture is shown in Figure 1.

---

**Algorithm 1** RISTA for Elastic DL Layer

**Input:** input signal $x$, kernel $\mathbf{A}$,
Initialize $z_0 \leftarrow \mathcal{A}(x)$
**for** $t = 1$ **to** $T - 1$ **do**
    $w_t \leftarrow \frac{1}{2|x - \mathcal{A}^*(z_t)|}$
    $r_t \leftarrow (\beta\mathbf{1} + (1 - \beta)w_t) \odot (x - \mathcal{A}^*(z_t))$
    $z_{t+1} \leftarrow \mathcal{T}_{\lambda t}(z_t + t \cdot \mathcal{A}(r_t))$
**end for**
**Output:** sparse code $z_T$

---

## 5 Experiment

In this section, we comprehensively evaluate the effectiveness of our proposed EDLNets under various experimental settings. Additionally, we provide several ablation studies to demonstrate the working mechanism of our approach.

### 5.1 Experimental Setting

**Datasets.** We conduct the experiments on several datasets including CIFAR10 (Krizhevsky et al., 2009), CIFAR100 (Krizhevsky et al., 2009) and Tiny-ImageNet (Le & Yang, 2015).

**Backbone architectures.** We select ResNets as the backbones, including ResNet10, ResNet18, ResNet34, and ResNet50 (He et al., 2016). Each of the convolutional layers in ResNets are replaced with our Elastic DL layer, resulting in the corresponding EDLNets. We use ResNet18 as the default backbone if not being specified.

**Evaluation methods.** We evaluate the performance of the models against various attacks, including FGSM (Goodfellow et al., 2014), PGD (Madry, 2017), C&W (Carlini & Wagner, 2017), AutoAttack (Croce & Hein, 2020), and SparseFool (Modas et al., 2019), covering budget measurements across $\ell_\infty$-norm, $\ell_2$-norm, and $\ell_1$-norm. For the PGD attack, we consider both $\ell_\infty$-norm and $\ell_2$-norm, denoted as PGD-$\ell_\infty$ and PGD-$\ell_2$, respectively. SparseFool uses the $\ell_1$-norm. Unless otherwise specified, $\ell_\infty$ is used as the default measurement. To prevent a false sense of security caused by gradient obfuscation, we perform multiple robustness reliability tests, including *certifiable robustness* (Figure 6), *transferability analysis* (Figure 7), and *zero-order gradient analysis* (Appendix D.3.4).

**Baselines.** For robust overfitting mitigation, we include the baselines including regularization ($\ell_1$, $\ell_2$ regularizations and their combination), Cutout (DeVries, 2017), Mixup (Zhang, 2017), and early stopping (Rice et al., 2020). For adversarial training methods, we compare the baselines including PGD-AT (Madry, 2017), TRADES (Zhang et al., 2019), MART (Wang et al., 2019), SAT (Huang et al., 2020), AWP (Wu et al., 2020), Consistency (Tack et al., 2022), DYNAT (Liu et al., 2024), PORT (Sehwag et al., 2021), and HAT (Rade & Moosavi-Dezfooli, 2022).

**Hyperparameter setting.** We train the baselines for 200 epochs with batch size 128, weight decay 2e-5, momentum 0.9, and an initial learning rate of 0.1 that is divided by 10 at the 100-th and 150-th epoch. For our Elastic DL, we pretrain the Vanilla DL model for 150 epochs and then fine-tune the Elastic DL model for 50 epochs.

## 5.2 ADVERSARIAL ROBUSTNESS & GENERALIZATION

First, we validate the effectiveness of our approach in mitigating overfitting. Next, we conduct a comprehensive evaluation of the adversarial training methods. Finally, we demonstrate our approach surpasses the state-of-the-art methods on the leaderboard by incorporating structural priors.

Table 3: Natural and robust performance of PGD-based adversarial training with different methods to mitigate the overfitting. BEST represents the highest test accuracy achieved during training, while FINAL is the average accuracy over the last five epochs. DIFF, the difference between BEST and FINAL, measures the ability to mitigate overfitting. The best performance is highlighted in **bold**, while the second-best is underlined.

| | NATURAL ACC. | | | ROBUST ACC. | | |
|---|---|---|---|---|---|---|
| **METHOD** | **FINAL** | **BEST** | **DIFF** | **FINAL** | **BEST** | **DIFF** |
| VANILLA | 78.98 | 79.90 | 0.92 | 44.90 | 48.01 | 3.11 |
| $\ell_1$ REG. | 64.84 | 65.71 | 0.87 | 40.94 | 41.97 | 1.03 |
| $\ell_2$ REG. | 78.88 | 79.39 | 0.51 | 42.73 | 48.26 | 5.53 |
| $\ell_2 + \ell_1$ REG. | 66.86 | 67.62 | 0.76 | 42.53 | 43.33 | 0.80 |
| CUTOUT | 75.11 | 75.58 | 0.47 | 47.12 | 48.23 | 1.11 |
| MIXUP | 69.64 | 72.05 | 2.41 | 46.10 | 48.53 | 2.43 |
| EARLY STOPPING | 75.51 | 75.51 | **0.00** | 47.69 | 47.95 | **0.26** |
| VANILLA DL | 82.59 | **83.27** | 0.68 | 44.03 | 50.53 | 6.50 |
| ELASTIC DL (OURS) | **83.01** | **83.27** | 0.26 | **54.94** | **55.66** | 0.72 |

**Robust overfitting mitigation.** To validate the effectiveness of incorporating structural priors, we compare our method with existing popular baselines in mitigating the *robust overfitting* problem in Table 3 and Figure 2. We leave the training curves of all the methods in Appendix D.2.1 and Appendix D.2.2 due to the space limit. From the results, we can make the following observations:

- From Table 3, we observe that our Elastic DL method not only achieves a significant advantage in both absolute FINAL and BEST performance but also maintains a relatively small gap (DIFF) between them, indicating that incorporating the structural prior effectively guides adversarial training to achieve better robustness and generalization.

- From Figure 2, we observe that during the 100th to 200th epochs, the Vanilla DL model exhibits a severe *robust overfitting* phenomenon. By incorporating our Elastic DL structural prior at the 150th

epoch, the test robustness improves substantially, highlighting the promising potential of the Elastic DL structural prior in overcoming the bottleneck of adversarial robustness and generalization.

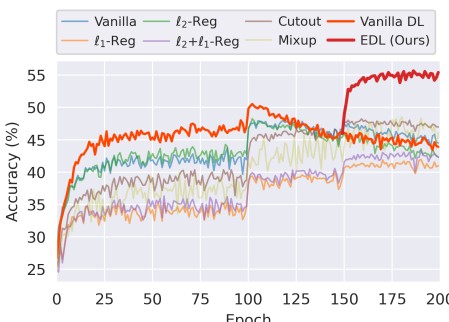

Figure 2: Test robust accuracy during the adversarial training. we pretrain the Vanilla DL model for 150 epochs and fine-tune the Elastic DL model starting from 150-th epoch. Our Elastic DL method can achieve the best adversarial robustness.

Table 4: Adversarial robsustness on CIFAR10 with ResNet18 as backbone. The best performance is highlighted in **bold**.

| METHOD | CLEAN | PGD | FGSM | C&W | AA |
|---|---|---|---|---|---|
| MART | 83.07 | 53.20 | 59.86 | 52.45 | 43.88 |
| SAT | 63.28 | 43.57 | 50.13 | 47.47 | 39.72 |
| AWP | 81.20 | 51.60 | 55.30 | 48.00 | 46.90 |
| CONSISTENCY | 84.37 | 45.19 | 53.84 | 43.75 | 40.88 |
| DYNAT | 82.34 | 52.25 | 65.96 | 52.19 | 45.10 |
| PGD-AT | 80.90 | 44.35 | 58.41 | 46.72 | 42.14 |
| + VANILLA DL | 83.28 | 45.64 | 53.88 | 41.22 | 43.70 |
| + ELASTIC (OURS) | 83.57 | 53.22 | 69.35 | 60.80 | 52.90 |
| TRADES-2.0 | 82.80 | 48.32 | 51.67 | 40.65 | 36.40 |
| + VANILLA DL | 79.05 | 40.64 | 47.12 | 41.49 | 34.90 |
| + ELASTIC (OURS) | 79.85 | 49.32 | 58.68 | 49.47 | 47.20 |
| TRADES-0.2 | 85.74 | 32.63 | 44.26 | 26.70 | 19.00 |
| + VANILLA DL | 82.55 | 25.37 | 44.48 | 30.3 | 15.30 |
| + ELASTIC (OURS) | 84.75 | 33.61 | 57.86 | 40.68 | 28.10 |
| PORT | 84.59 | 58.62 | 62.64 | 58.12 | 55.14 |
| + VANILLA DL | 82.35 | 56.40 | 60.68 | 56.77 | 54.00 |
| + ELASTIC (OURS) | 82.76 | 59.00 | 68.54 | 60.92 | 56.30 |
| HAT | 85.95 | 56.29 | 61.17 | 49.52 | 53.16 |
| + VANILLA DL | 86.42 | 57.79 | 62.67 | 51.61 | 54.30 |
| + ELASTIC (OURS) | **86.84** | **62.48** | **71.46** | **59.90** | **59.07** |

**Adversarial training robustness.** To validate the effectiveness of our Elastic DL, we select several existing popular adversarial defenses and report the experimental results of backbone ResNet18 under various attacks in Table 4. From the results we can make the following observations:

- Our HAT + Elastic DL significantly outperforms other methods across various attacks, achieving state-of-the-art performance among all baselines.

- Our Elastic DL is a robust architecture that is orthogonal to existing adversarial training methods and can be combined with them to further improve robustness.

**SOTA performance on leaderboard.** Furthermore, we validate whether incorporating our structural prior improves over state-of-the-art methods. To achieve this, we select the top-ranking methods, HAT (Rade & Moosavi-Dezfooli, 2022) and PORT (Sehwag et al., 2021)), listed on the Robust-Bench (Croce et al., 2020) leaderboard under $\ell_\infty$-norm and $\ell_2$-norm attacks, using ResNet-18 on the CIFAR-10 dataset. As shown in Table 5 ($\ell_\infty$-norm attack) and Table 6 ($\ell_2$-norm attack), Our methods, HAT+Elastic DL and PORT+Elastic DL, consistently achieve superior performance in most cases for both natural and robust performances.

Table 5: State-of-the-art performance of ResNet18 on CIFAR10 under $\ell_\infty$-norm attack.

| LEADERBOARD UNDER $\ell_\infty$-NORM ATTACK | | | | | | |
|---|---|---|---|---|---|---|
| | CLEAN | PGD-$\ell_\infty$ | | | AUTOATTACK-$\ell_\infty$ | | |
| BUDGET | 0 | $\frac{8}{255}$ | $\frac{16}{255}$ | $\frac{32}{255}$ | $\frac{8}{255}$ | $\frac{16}{255}$ | $\frac{32}{255}$ |
| PORT | 84.59 | 58.62 | 27.49 | 5.79 | 55.14 | 17.8 | 0.3 |
| + VANILLA DL | 82.35 | 56.4 | 27.3 | 6.38 | 54.0 | 20.4 | 0.6 |
| + ELASTIC (OURS) | 82.76 | 59.0 | 36.53 | 22.17 | 56.3 | 24.6 | 1.7 |
| HAT | 85.95 | 56.29 | 25.82 | 6.09 | 53.16 | 17.20 | 0.60 |
| + VANILLA DL | 86.42 | 57.79 | 26.08 | 6.07 | 54.30 | 17.56 | 0.63 |
| + ELASTIC (OURS) | 86.84 | 62.48 | 44.66 | 33.69 | 59.10 | 29.93 | 2.10 |

Table 6: State-of-the-art performance of ResNet18 on CIFAR10 under $\ell_2$-norm attack.

| LEADERBOARD UNDER $\ell_2$-NORM ATTACK | | | | | | |
|---|---|---|---|---|---|---|
| | CLEAN | PGD-$\ell_2$ | | | AUTOATTACK-$\ell_2$ | | |
| BUDGET | 0 | 0.5 | 1.0 | 2.0 | 0.5 | 1.0 | 2.0 |
| PORT | 88.82 | 74.89 | 54.47 | 27.69 | **73.80** | 48.1 | 5.90 |
| + VANILLA DL | 87.34 | 73.52 | 53.75 | 27.5 | 71.8 | 49.1 | 6.7 |
| + ELASTIC (OURS) | 87.81 | **75.56** | **60.76** | **41.44** | 72.2 | **52.4** | **11.1** |
| HAT | 89.92 | 74.68 | 47.67 | 21.38 | 72.9 | 40.8 | 2.2 |
| + VANILLA DL | 88.84 | 67.99 | 40.87 | 17.97 | 66.8 | 27.8 | 0.6 |
| + ELASTIC (OURS) | **89.95** | 74.62 | 51.41 | 27.05 | 73.2 | 44.5 | 3.2 |

## 5.3 ABLATION STUDY

**Universality across datasets and backbones.** To validate the consistent effectiveness of our proposed methods, we conduct comprehensive abation studies on the different backbones (ResNet10, ResNet18, ResNet34, ResNet50), datasets (CIFAR10, CIFAR100, Tiny-ImageNet). As demonstrated in the Figure 3, Table 8, 9 and 10 in Appendix D.3.1, our proposed Elastic DL exhibit excellent clean performance and robustness under various attacks.

**Hidden embedding visualization.** We also conduct visualization analyses on the hidden embedding to obtain better insight into the effectiveness of our proposed Elastic DL. We begin by quantifying the relative difference between clean embeddings ($x$ or $z_i$) and attacked embeddings ($x'$ or $z_i'$) across all

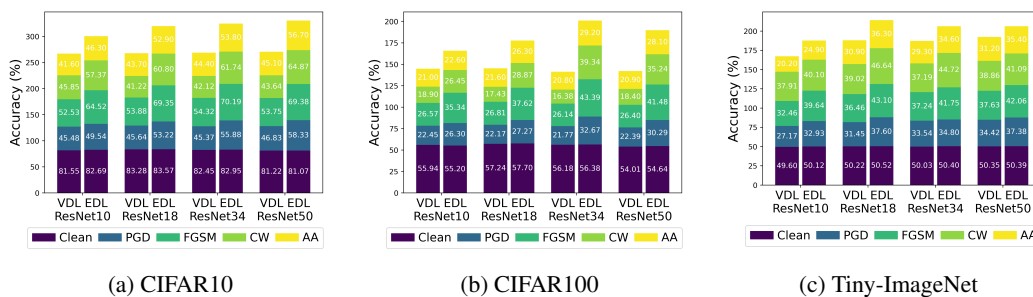

(a) CIFAR10 (b) CIFAR100 (c) Tiny-ImageNet

Figure 3: Adversarial robustness under various settings. Our Elastic DL outperforms Vanilla DL across various datasets (CIFAR10 / CIFAR100 / Tiny-ImageNet), backbones (ResNet10 / ResNet18 / ResNet34 / ResNet50) and attacks (PGD / FGSM / CW / AA).

layers, as shown in Figure 5. Additionally, we visualize one instance in Figure 4, with more examples provided in Appendix D.3.5. The results in Figure 5 show that Elastic DL has smaller embedding difference across layers, indicating that our proposed Elastic DL architecture indeed mitigates the impact of the adversarial perturbation.

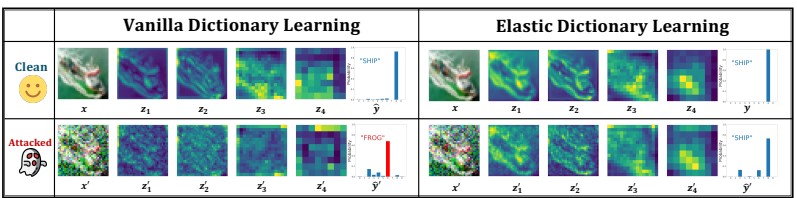

Figure 4: Hidden embedding visualization . The difference between clean and attacked embeddings in Elastic DL is smaller compared to Vanilla DL, with this effect becoming more significant in deeper layers. Consequently, while an adversarial attack alters the Vanilla DL output from "*SHIP*" to "*FROG*", Elastic DL successfully preserves the correct prediction.

Figure 5: Embedding difference. Our Elastic DL shows smaller embedding difference than Vanilla DL.

**Certifiable robustness.** We also provide the results of certifiable robustness via randomized smoothing, which is a certified defense that can theoretically guarantee certified accuracy regardless of the evaluated attacks. The results in Figure 6 demonstrate that our Elastic DL delivers better certified robustness than the vanilla DL.

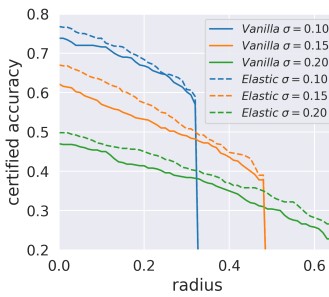

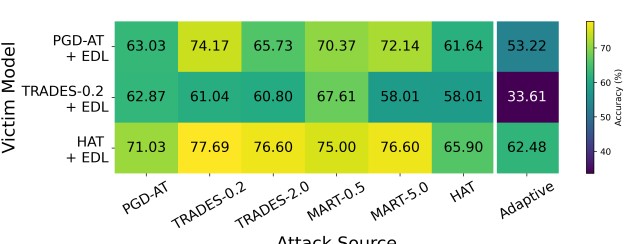

Figure 6: Certifiable robustness. Elastic DL delivers better certified robustness than vanilla DL.

Figure 7: Transferability analysis. We evaluate the transfer attacks from multiple baselines, using the adaptive attack as a comparison, where the adaptive attack demonstrates the strongest performance.

**Transferability analysis.** To validate the effectiveness of our method and strength of evaluated adaptive attack, we evaluate transfer attacks from multiple baselines, along with the adaptive attack for comparison. It can be observed from Figure 7 that the adaptive attack yields the strongest attack, thereby validating the effectiveness of our experimental evaluation.

**Different attack measurements.** In addition to $\ell_\infty$-norm attack (PGD-$\ell_\infty$), we also validate the consistent effectiveness of our Elastic DL with $\ell_2$-norm (PGD-$\ell_2$) and $\ell_1$-norm (SparseFool) attacks in the Figure 18 and Table 11 in Appendix D.3.3.

**Convergence.** To validate the effectiveness of our RISTA iterations, we plot the loss descent curves of overall objective Eq.(6) along with the individual terms ($\|\boldsymbol{x} - \mathcal{A}^*(\boldsymbol{z})\|_2^2$, $\|\boldsymbol{x} - \mathcal{A}^*(\boldsymbol{z})\|_1$ and $\|\boldsymbol{z}\|_1$) in Figure 8, which shows that RISTA converges rapidly within first three steps.

**Attack behavior.** To investigate the attack behaviors, we apply the PGD attack to both models and visualize the perturbations in Figure 9. It can be observed that, in the Vanilla DL, the adversarial attack introduces substantial outlying noise, which can be largely mitigated by our Elastic DL.

**Out-of-distribution robustness.** Beyond in-distribution robustness, we further validate the advantage of our proposed Elastic DL structure by evaluating the out-of-distribution performance of Vanilla DL and Elastic DL. The results in Figure 10 demonstrate the superiority of our Elastic DL over the Vanilla DL under various types of out-of-distribution noise.

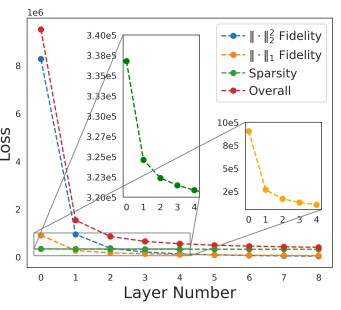 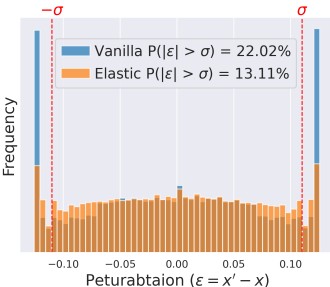 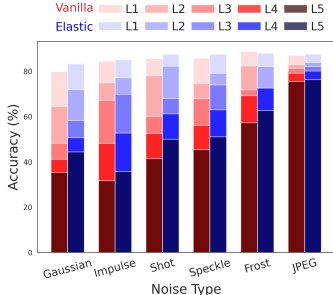

Figure 8: Algorithm convergence. RISTA algorithm achieves fast convergence within just three steps.

Figure 9: Attack behaviors. The attacker tends to attack Vanilla DL model by introducing outlying values.

Figure 10: Out-of-distribution robustness. Our Elastic DL also demonstrates excellent out-of-distribution robustness.

**Running time analysis.** We also perform an analysis to evaluate the inference time of different architectures using ResNet18 as the backbone. We replace multiple convolutional layers in ResNet18 with either Vanilla DL or Elastic DL layers, rang-

Table 7: Running time (ms) analysis.

| LAYERS | 0 (RESNET) | 2 | 4 | 6 | 8 | 10 | 12 | 14 |
|---|---|---|---|---|---|---|---|---|
| VANILLA DL | 7.82 | 8.40 | 9.28 | 10.51 | 12.13 | 13.11 | 14.16 | 15.40 |
| ELASTIC DL | 7.82 | 8.90 | 11.39 | 13.18 | 15.99 | 16.86 | 19.57 | 21.94 |

ing from 0 to 14 layers. As shown in Table 7, our Elastic DL introduces only a slight computational overhead compared to Vanilla DL and requires 1-3 times more computation than ResNets, which is considered acceptable. However, our Elastic DL demonstrates significantly improved robustness compared to ResNets and Vanilla DL.

## 6 CONCLUSION & LIMITATION

This paper proposes an orthogonal direction to break through the current plateau of adversarial robustness. We begin by revealing the vulnerability of dictionary learning in deep learning, and propose a novel elastic dictionary learning approach along with an efficient RISTA algorithm. Our comprehensive experiments demonstrate that our method achieves remarkable robustness, surpassing state-of-the-art baselines available on the robustness leaderboard. To the best of our knowledge, this is the first work to discover and validate that structural prior can reliably enhance adversarial robustness and generalization, unveiling a promising direction for future research.

Regarding the limitations, the efficiency can be further improved by either enhancing the algorithm or actively selecting the layers to be replaced. Additionally, we highlight a promising direction: while this work focuses solely on the dictionary learning prior, more diverse structural priors could be explored within the same paradigm in the future.

## 7 ETHICS STATEMENT

This paper investigates a dictionary learning-based robust architecture to enhance model robustness. We have not identified any ethical concerns related to human subjects, data release practices, conflicts of interest or sponsorship, discrimination, bias or fairness, or issues of research integrity.

## 8 REPRODUCIBILITY STATEMENT

We provide comprehensive details to facilitate the reproduction of our experiments. Specifically, the datasets, models, and attack methods are described in Section 5.1, along with the hyperparameters used in our proposed method. The code will be released upon paper acceptance.

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

# A   OVERVIEW OF ELASTIC DICTIONARY LEARNING

**Overview of Elastic DL neural networks.** Here we plot a figure to show the overall pipeline of incorporating Elastic DL structural prior into adversarial training as in Figure 11.

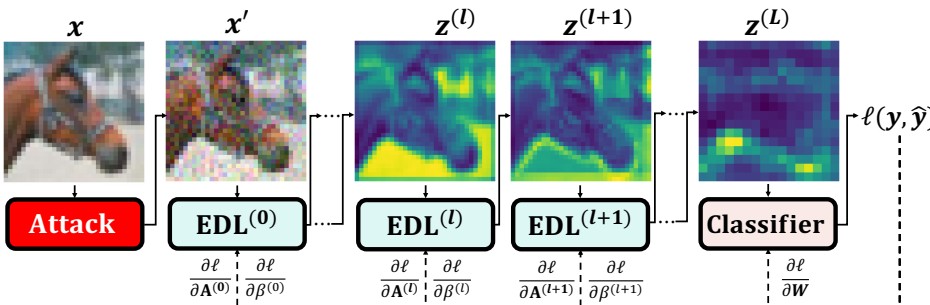

Figure 11: Overview of Elastic DL neural networks in adversarial training. Elastic DL neural networks consist of multiple stacked Elastic DL (EDL) layers. During the forward pass, the input $x$ is fed into the model, generating a series of hidden codes $\{z^{(l)}\}_{l=1}^{L}$ through EDL layers. During the backward pass, the model parameters are updated, including kernel weights $\{\mathbf{A}^{(l)}\}_{l=0}^{L-1}$, layer-wise balance weights $\{\beta^{(l)}\}_{l=0}^{L-1}$, and classifier parameters $\mathbf{W}$.

Consider a model with $\{\mathbf{A}^{(l)}\}_{l=0}^{L-1}$ and $\{\beta^{(l)}\}_{l=0}^{L-1}$ in the $L$ EDL layers and $\mathbf{W}$ in the classifier. Then, the adversarial training framework with EDL can be formulated as:

$$\min_{\{\mathbf{A}^{(l)},\beta^{(l)}\}_{l=0}^{L-1}} \mathbb{E}_{(x,y)\sim\mathcal{D}} \left[ \max_{x'\in\mathcal{B}(x)} \ell(z^{*(L)}, y) \right]$$

$$\text{s.t. } z^{*(l+1)} = \arg\min_{z} \ell_{NADL}(z, \mathbf{A}^{(l)}, z^{*(l)}),$$

$$\ell_{NADL}^{(l)}(z, \mathbf{A}, x) \text{ is defined in Eq. (6),}$$

$$z^{*(0)} = x',$$

$$\text{for } l = 0, \cdots, L-1.$$

Its overall pipeline can be divided into three main steps as in Figure 11:

- Step 1 (Attack): leverage adversarial attack algorithm (e.g., PGD) to generate worst-case perturbation $x'$.

- Step 2 (Forward): input $x'$ as $z^{*(0)}$ into model to obtain a series of hidden codes for each layer $\{z^{(l)}\}_{l=1}^{L}$ by optimizing dictionary learning loss in Eq. (6).

- Step 3 (Backward): update the model parameters including kernel weights $\{\mathbf{A}^{(l)}\}_{l=0}^{L-1}$, layer-wise balance weight $\{\beta^{(l)}\}_{l=0}^{L-1}$, and other parameters $\mathbf{W}$.

**Exploded view of Elastic DL layer.** We also provide an exploded view of each Elastic DL layer as in Figure 12. The input signal $z^{(k)}$ can be represented by a linear superposition of several atoms $\{\alpha_{dc}\}$ from a convolutional dictionary $\mathbf{A}^{(l)}$. Each EDL layer is unrolled using the proposed RISTA algorithm, which approximates the solution for elastic dictionary learning objective.

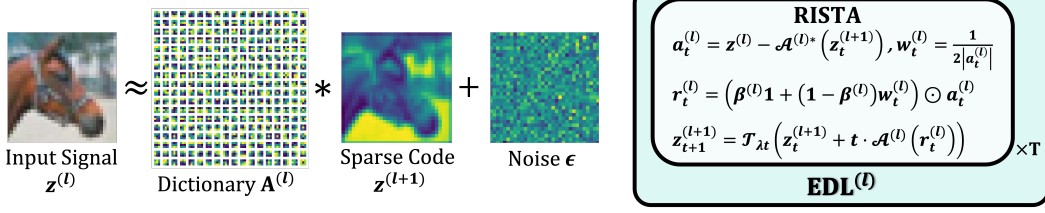

Input Signal $z^{(l)}$   Dictionary $\mathbf{A}^{(l)}$   Sparse Code $z^{(l+1)}$   Noise $\epsilon$

**RISTA**
$$a_t^{(l)} = z^{(l)} - \mathcal{A}^{(l)*}\left(z_t^{(l+1)}\right), w_t^{(l)} = \frac{1}{2|a_t^{(l)}|}$$
$$r_t^{(l)} = \left(\beta^{(l)}\mathbf{1} + (1-\beta^{(l)})w_t^{(l)}\right) \odot a_t^{(l)}$$
$$z_{t+1}^{(l+1)} = \mathcal{T}_{\lambda t}\left(z_t^{(l+1)} + t \cdot \mathcal{A}^{(l)}\left(r_t^{(l)}\right)\right) \times T$$
**EDL$^{(l)}$**

Figure 12: Exploded view of Elastic DL (EDL) layer.

# B THEORETICAL PROOF

## B.1 PROOF OF LEMMA 4.1

*Proof.* Since $\sqrt{a} \le \frac{a}{2\sqrt{b}} + \frac{\sqrt{b}}{2}$ and the equlity holds when $a = b$, by replacemnet as $a = (\boldsymbol{x}[i,j,c] - \mathcal{A}^*(\boldsymbol{z})[i,j,c])^2$ and $b = (\boldsymbol{x}[i,j,c] - \mathcal{A}^*(\boldsymbol{z}_*)[i,j,c])^2$, then

$$|\boldsymbol{x}[i,j,c] - \mathcal{A}^*(\boldsymbol{z})[i,j,c]| \le \frac{1}{2} \cdot \frac{1}{|\boldsymbol{x}[i,j,c] - \mathcal{A}^*(\boldsymbol{z}_*)[i,j,c]|} \cdot (\boldsymbol{x}[i,j,c] - \mathcal{A}^*(\boldsymbol{z})[i,j,c])^2 + \frac{1}{2}|\boldsymbol{x}[i,j,c] - \mathcal{A}^*(\boldsymbol{z}_*)[i,j,c]|$$

$$= \boldsymbol{w}[i,j,c] \cdot (\boldsymbol{x}[i,j,c] - \mathcal{A}^*(\boldsymbol{z})[i,j,c])^2 + \frac{1}{2}|\boldsymbol{x}[i,j,c] - \mathcal{A}^*(\boldsymbol{z}_*)[i,j,c]|$$

Sum up the items on both sides, we obtain

$$\mathcal{R}(\boldsymbol{z}) = \|\boldsymbol{x} - \mathcal{A}^*(\boldsymbol{z})\|_1 = \sum_{i,j,c} |\boldsymbol{x}[i,j,c] - \mathcal{A}^*(\boldsymbol{z})[i,j,c]|$$

$$\le \sum_{i,j,c} \boldsymbol{w}[i,j,c] \cdot (\boldsymbol{x}[i,j,c] - \mathcal{A}^*(\boldsymbol{z})[i,j,c])^2 + \frac{1}{2} \sum_{i,j,c} |\boldsymbol{x}[i,j,c] - \mathcal{A}^*(\boldsymbol{z}_*)[i,j,c]|$$

$$= \|\boldsymbol{w}^{1/2} \odot (\boldsymbol{x} - \mathcal{A}^*(\boldsymbol{z}))\|_2^2 + \frac{1}{2}\mathcal{R}(\boldsymbol{z}_*)$$

$$= \mathcal{U}(\boldsymbol{z}, \boldsymbol{z}_*)$$

and the equality holds at $a = b$ ($\boldsymbol{z} = \boldsymbol{z}_*$):

$$\mathcal{U}(\boldsymbol{z}_*, \boldsymbol{z}_*) = \mathcal{R}(\boldsymbol{z}_*). \tag{7}$$

$\square$

## B.2 PROOF OF ALGORITHM ITERATION IN EQ. (5)

Here, we derive the algorithm for general elastic dictionary learning (Elastic DL), the $\ell_1$-based robust dictionary learning (Robust DL) can be consider as the special case with $\beta = 0$.

*Proof.* For convex objective:

$$f(\boldsymbol{z}) = \frac{\beta}{2}\|\boldsymbol{x} - \mathcal{A}^*(\boldsymbol{z})\|_2^2 + \frac{1-\beta}{2}\|(\boldsymbol{w}^{(t)})^{1/2} \odot (\boldsymbol{x} - \mathcal{A}^*(\boldsymbol{z}))\|_2^2,$$

we can achieve the optima via the first-order gradient descent:

$$\boldsymbol{z}_{t+1} = \boldsymbol{z}_t - t\nabla f(\boldsymbol{z}_t),$$

or equivalently,

$$\boldsymbol{z}_{t+1} = \arg\min_{\boldsymbol{z}}\{f(\boldsymbol{z}_t) + \langle \boldsymbol{z} - \boldsymbol{z}_t, \nabla f(\boldsymbol{z}_t)\rangle + \frac{1}{2t}\|\boldsymbol{z} - \boldsymbol{z}_t\|^2\}.$$

Then, for the corresponding $\ell_1$-regularized problem:

$$\min_{\boldsymbol{z}} f(\boldsymbol{z}) + \lambda\|\boldsymbol{z}\|_1,$$

we have:

$$\boldsymbol{z}_{t+1} = \arg\min_{\boldsymbol{z}}\{f(\boldsymbol{z}_t) + \langle \boldsymbol{z} - \boldsymbol{z}_t, \nabla f(\boldsymbol{z}_t)\rangle + \frac{1}{2t}\|\boldsymbol{z} - \boldsymbol{z}_t\|^2 + \lambda\|\boldsymbol{z}\|_1\}$$

$$= \arg\min_{\boldsymbol{z}}\{\frac{1}{2t}\|\boldsymbol{z} - (\boldsymbol{z}_t - t\nabla f(\boldsymbol{z}_t))\|^2 + \lambda\|\boldsymbol{z}\|_1\}$$

$$= \arg\min_{\boldsymbol{z}}\{g(\boldsymbol{z}) := \frac{1}{2t}\|\boldsymbol{z} - \boldsymbol{y}\|^2 + \lambda\|\boldsymbol{z}\|_1\} \quad (\boldsymbol{y} = \boldsymbol{z}_t - t\nabla f(\boldsymbol{z}_t))$$

Then, the optimality condition is:

$$0 \in \partial_{\boldsymbol{z}} g(\boldsymbol{z}^*) = \frac{1}{t}(\boldsymbol{z}^* - \boldsymbol{y}) + \lambda \mathrm{sign}(\boldsymbol{z}^*)$$

$$\Leftrightarrow \quad \boldsymbol{y} \in \boldsymbol{z}^* + \lambda t \mathrm{sign}(\boldsymbol{z}^*)$$

$$\Leftrightarrow \quad \boldsymbol{y} \in (\mathrm{Id} + \lambda t \mathrm{sign}(\cdot))(\boldsymbol{z}^*)$$

$$\Leftrightarrow \quad \boldsymbol{z}^* = \mathcal{T}_{\lambda t}(\boldsymbol{y}) := (\mathrm{Id} + \lambda t \mathrm{sign}(\cdot))^{-1}(\boldsymbol{y}) = \mathrm{sign}(\boldsymbol{y})(|\boldsymbol{y} - \lambda t|)_+ .$$

Since

$$\nabla f(\boldsymbol{z}) = -\beta \mathcal{A}(\boldsymbol{x} - \mathcal{A}^*(\boldsymbol{z})) - (1 - \beta)\mathcal{A}(\boldsymbol{w}^{(t)} \odot (\boldsymbol{x} - \mathcal{A}^*(\boldsymbol{z})))$$

$$= -\mathcal{A}\left(\left(\beta \mathbf{1} + (1 - \beta)\boldsymbol{w}^{(t)}\right) \odot (\boldsymbol{x} - \mathcal{A}^*(\boldsymbol{z}_t))\right),$$

Then

$$\boldsymbol{z}_{t+1} = \boldsymbol{z}^* = \mathcal{T}_{\lambda t}(\boldsymbol{y}) = \mathcal{T}_{\lambda t}(\boldsymbol{z}_t - t \cdot \nabla f(\boldsymbol{z})) = \mathcal{T}_{\lambda t}\left(\boldsymbol{z}_t + t \cdot \mathcal{A}\left(\left(\beta \mathbf{1} + (1 - \beta)\boldsymbol{w}^{(t)}\right) \odot (\boldsymbol{x} - \mathcal{A}^*(\boldsymbol{z}_t))\right)\right)$$

$\square$

## C RELATED WORKS

### C.1 ADVERSARIAL ATTACKS

Adversarial attacks are typically classified into two main categories: *white-box* and *black-box* attacks. In white-box attacks, the attacker has full knowledge of the target neural network, including its architecture, parameters, and gradients. Common examples of white-box attacks include gradient-based methods such as FGSM (Goodfellow et al., 2014), DeepFool (Moosavi-Dezfooli et al., 2016), PGD (Madry, 2017), and the C&W attack (Carlini & Wagner, 2017). In contrast, black-box attacks operate under limited information, where the attacker can only interact with the model through its input-output behavior without direct access to internal details. Examples of black-box methods include surrogate model-based approaches (Papernot et al., 2017), zeroth-order optimization techniques (Chen et al., 2017), and query-based methods (Andriushchenko et al., 2020; Alzantot et al., 2019).

Here we list the detailed information of attacks we use in the main paper:

- Fast Gradient Sign Method (FGSM) (Goodfellow et al., 2014): FGSM is one of the earliest and most widely used adversarial attack methods. It generates adversarial examples by using the gradient of the loss function with respect to the input data to craft small but purposeful perturbations that lead the model to make incorrect predictions.

- Projected Gradient Descent (PGD) (Madry, 2017): PGD is an iterative and more robust extension of FGSM. It repeatedly applies small perturbations within a defined range (or epsilon ball) to maximize the model's loss. PGD is often considered a strong adversary in the evaluation of model robustness.

- Carlini & Wagner Attack (C&W) (Carlini & Wagner, 2017): This attack focuses on crafting adversarial examples by optimizing a custom loss function designed to minimize perturbations while ensuring the generated adversarial samples are misclassified.

- AutoAttack Croce & Hein (2020): AutoAttack is an ensemble of adversarial attack methods that automatically evaluates the robustness of models. It combines various attacks to provide a strong, reliable benchmark for adversarial robustness without manual tuning.

- SparseFool Modas et al. (2019): SparseFool is a sparse adversarial attack designed to generate adversarial examples by perturbing only a few pixels in the input image. It highlights how minimal changes can significantly alter model predictions.

### C.2 ADVERSARIAL DEFENSES

Significant efforts have been devoted to enhancing model robustness through a variety of strategies, including detection techniques (Metzen et al., 2017; Feinman et al., 2017; Grosse et al., 2017; Sehwag et al., 2021; Rade & Moosavi-Dezfooli, 2022; Addepalli et al., 2022), purification-based approaches (Ho & Vasconcelos, 2022; Nie et al., 2022; Shi et al., 2021; Yoon et al., 2021), robust training methods (Madry, 2017; Zhang et al., 2019; Gowal et al., 2021; Li & Liu, 2023), and regularization-based techniques (Cisse et al., 2017; Zheng et al., 2016). Among these, adversarial training-based methods (Sehwag et al., 2021; Rade & Moosavi-Dezfooli, 2022; Addepalli et al., 2022) have proven highly effective against adaptive adversarial attacks, consistently leading the robustness leaderboard (RobustBench) (Croce et al., 2020). Despite their success, most existing methods rely heavily on extensive synthetic training data generated by advanced models, larger network architectures, and empirically driven training strategies. These dependencies pose substantial challenges to advancing beyond the current plateau in adversarial robustness. In this work, we introduce an elastic dictionary framework that incorporates structural priors into model design. This approach is fully orthogonal to existing methods and offers a complementary pathway to further enhance robustness when integrated with current techniques.

Here are we list the detailed information of adversarial training based methods we use in the main paper:

- PGD-AT (Madry, 2017): Projected Gradient Descent Adversarial Training (PGD-AT) is a fundamental adversarial training approach that enhances model robustness by iteratively generating adversarial examples using PGD and training the model on them.

- TRADES (Zhang et al., 2019): TRADES (Tradeoff-inspired Adversarial Defense via Surrogate Loss Minimization) balances robustness and accuracy by introducing a regularization term that penalizes the discrepancy between natural and adversarial predictions.

- MART (Wang et al., 2019): Misclassification-Aware Adversarial Training (MART) improves robustness by assigning higher weights to misclassified examples, emphasizing correctly classified samples' robustness.

- SAT (Huang et al., 2020): Self-Adaptive Training (SAT) refines adversarial training by adjusting the training process based on the model's confidence, mitigating the effects of incorrect labels and improving generalization.

- AWP (Wu et al., 2020): Adversarial Weight Perturbation (AWP) enhances robustness by perturbing model parameters within a constrained space to improve the worst-case performance against adversarial attacks.

- Consistency (Tack et al., 2022): Consistency training leverages perturbation-invariant representations to enhance robustness by enforcing consistent predictions across different transformations of inputs.

- DYNAT (Liu et al., 2024): Dynamic Adversarial Training (DYNAT) adapts training strategies dynamically based on model performance, balancing robustness and generalization efficiency.

- PORT (Sehwag et al., 2021): Proxy Distribution-based Robust Training (PORT) leverages data from proxy distributions, such as those generated by advanced generative models, to enhance adversarial robustness. By formally analyzing robustness transfer and optimizing training, PORT demonstrates significant improvements in robustness under various threat models.

- HAT (Rade & Moosavi-Dezfooli, 2022): Helper-based Adversarial Training (HAT) mitigates the accuracy-robustness trade-off by incorporating additional incorrectly labeled examples during training. This approach reduces excessive margin changes along certain adversarial directions, improving accuracy without compromising robustness and achieving a better trade-off compared to existing methods.

# D  ADDITIONAL EXPERIMENTS

## D.1  PRELIMINARY STUDIES

**Preliminary in SDNet18.** We evaluate Vanilla DL (with both fixed and tuned $\lambda$) under random impulse noise and adaptive PGD adversarial attacks Madry (2017). In our experiments, the noise level corresponds to the noise density $c$, i.e., the proportion of pixels in the image that are randomly replaced with either the minimum or maximum pixel value. Specifically, we set the noise levels as follows: L-1 ($c = 0.03$), L-2 ($c = 0.06$), L-3 ($c = 0.09$), L-4 ($c = 0.12$), and L-5 ($c = 0.15$). As illustrated in Figure 13, increasing the noise level and intensifying the distribution tail degradation lead to a decline in Vanilla DL's accuracy. While tuning the sparsity weight $\lambda$ enhances resilience to random noise, models with any $\lambda$ suffer a sharp performance drop under adaptive PGD attacks, with accuracy nearing zero.

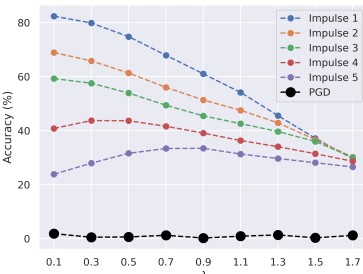

Figure 13: Performance of SDNet18 (Vanilla DL) under random Impulse noise with different levels.

## D.2 ADVERSARIAL TRAINING CURVES

### D.2.1 TRAINING CURVES OF EACH METHOD

**Training curve of our Elastic DL.** From Figure 14, we can observe that during the 100th - 150th epochs, the Vanilla DL model exhibits a severe *robust overfitting* phenomenon: while training performance improves, the test robust accuracy drops significantly. After incorporating our Elastic DL structural prior at the 150th epoch, both training and testing robustness improve substantially. Although there is a slight drop in natural performance during the initial switching period, it recovers quickly within a few epochs. This phenomenon highlights the promising potential of the Elastic DL structural prior in breaking through the bottleneck of adversarial robustness and generalization.

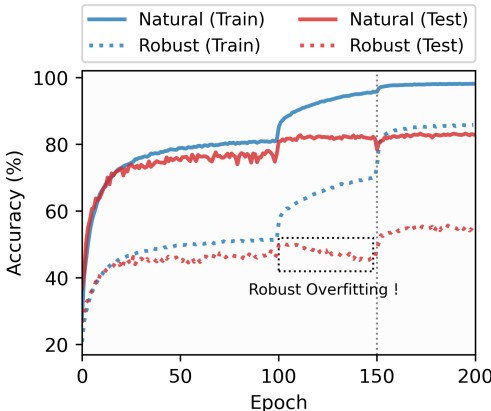

Figure 14: Adversarial training curve of our Elastic DL. During the 100th to 150th epochs, the model experiences a catastrophic *robust overfitting* problem. By introducing the Elastic DL structural prior at the 150th epoch and fine-tuning, we effectively mitigate overfitting and achieve significantly improved robustness and generalization.

**Training curves of baseline methods.** We track the training curves of the baselines including regularization ($\ell_1$, $\ell_2$ regularizations and their combination), Cutout DeVries (2017), Mixup Zhang (2017) in Figure 15.

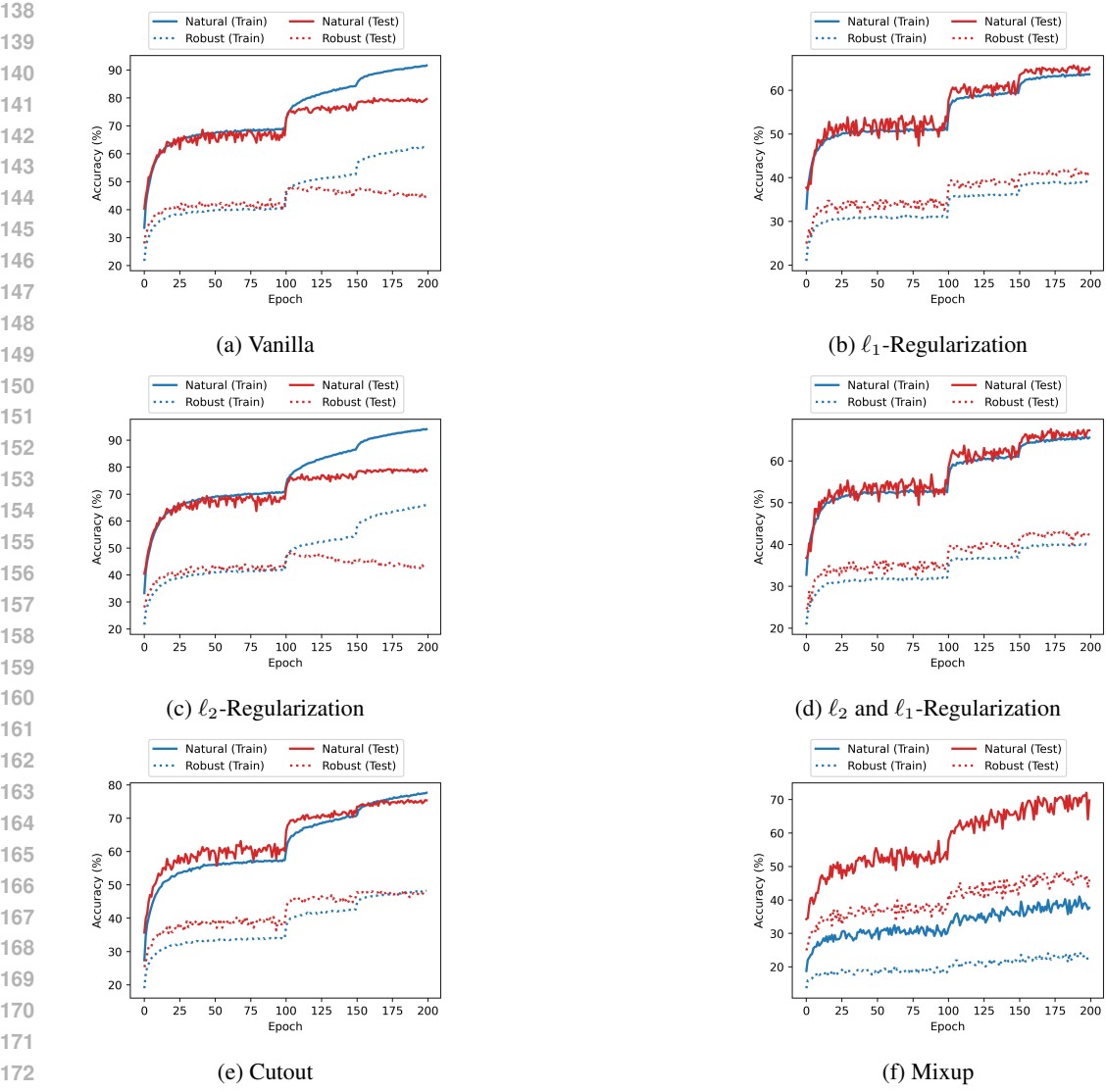

Figure 15: Training curves of baselines.

### D.2.2 COMPARISON OF ALL METHODS

To make a comparison of all the methods, we compare the natural and robust performance in the training and testing dataset through the training curve in Figure 16. The figures show the consisente advantage of our Elastic DL over other methods.

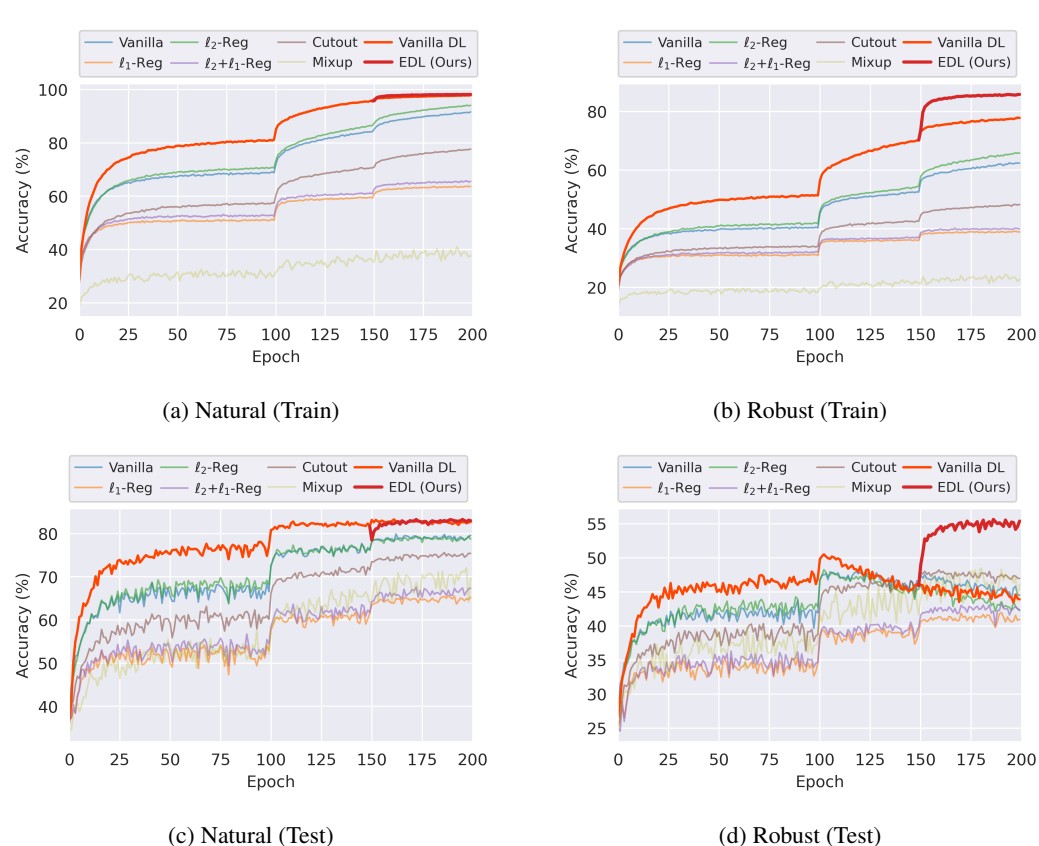

Figure 16: Comparison of training curves of all methods.

## D.3 ABLATION STUDIES

### D.3.1 UNIVERSALITY

**Universality across various backbones, datasets and attacks.** We conduct ablation studies on different backbones, datasets, and attacks in Table 8, Table 9, and Table 10. Our proposed method shows consistent effectiveness under various settings.

Table 8: Adversarial robsustness on CIFAR10 with different backbones.

| METHOD | NATURAL | PGD | FGSM | C&W | AA |
|---|---|---|---|---|---|
| VANILLA DL + RESNWT10 | 81.55 | 45.48 | 52.53 | 45.85 | 41.60 |
| ELASTIC DL + RESNET10 | 82.69 | 49.54 | 64.52 | 57.37 | 46.30 |
| VANILLA DL + RESNWT18 | 83.28 | 45.64 | 53.88 | 41.22 | 43.70 |
| ELASTIC DL + RESNET18 | 83.57 | 53.22 | 69.35 | 60.8 | 52.90 |
| VANILLA DL + RESNWT34 | 82.45 | 45.37 | 54.32 | 42.12 | 44.40 |
| ELASTIC DL + RESNET34 | 82.95 | 55.88 | 70.19 | 61.74 | 53.80 |
| VANILLA DL + RESNWT50 | 81.22 | 46.83 | 53.75 | 43.64 | 45.10 |
| ELASTIC DL + RESNET50 | 81.07 | 58.33 | 69.38 | 64.87 | 56.70 |

Table 9: Adversarial robsustness on CIFAR100 with different backbones.

| METHOD | NATURAL | PGD | FGSM | C&W | AA |
|---|---|---|---|---|---|
| VANILLA DL + RESNWT10 | 55.94 | 22.45 | 26.57 | 18.9 | 21.00 |
| ELASTIC DL + RESNET10 | 55.20 | 26.30 | 35.34 | 26.45 | 22.60 |
| VANILLA DL + RESNWT18 | 57.24 | 22.17 | 26.81 | 17.43 | 21.60 |
| ELASTIC DL + RESNET18 | 57.70 | 27.27 | 37.62 | 28.87 | 26.30 |
| VANILLA DL + RESNWT34 | 56.18 | 21.77 | 26.14 | 16.38 | 20.80 |
| ELASTIC DL + RESNET34 | 56.38 | 32.67 | 43.39 | 39.34 | 29.20 |
| VANILLA DL + RESNWT50 | 54.01 | 22.39 | 26.4 | 18.4 | 20.90 |
| ELASTIC DL + RESNET50 | 54.64 | 30.29 | 41.48 | 35.24 | 28.10 |

Table 10: Adversarial robsustness on Tiny-Imagenet with different backbones.

| METHOD | NATURAL | PGD | FGSM | C&W | AA |
|---|---|---|---|---|---|
| VANILLA DL + RESNWT10 | 49.6 | 27.17 | 32.46 | 37.91 | 20.20 |
| ELASTIC DL + RESNET10 | 50.12 | 32.93 | 39.64 | 40.10 | 24.90 |
| VANILLA DL + RESNWT18 | 50.22 | 31.45 | 36.46 | 39.02 | 30.90 |
| ELASTIC DL + RESNET18 | 50.52 | 37.6 | 43.1 | 46.64 | 36.30 A |
| VANILLA DL + RESNWT34 | 50.03 | 33.54 | 37.24 | 37.19 | 29.30 |
| ELASTIC DL + RESNET34 | 50.40 | 34.8 | 41.75 | 44.72 | 34.60 |
| VANILLA DL + RESNWT50 | 50.35 | 34.42 | 37.63 | 38.86 | 31.20 |
| ELASTIC DL + RESNET50 | 50.39 | 37.38 | 42.06 | 41.09 | 35.40 |

### D.3.2 ORTHOGONALITY TO ADVERSARIAL TRAINING.

Our proposed Elastic DL framework incorporates structural priors into neural networks, complementing existing adversarial training techniques. As shown in Table 4 and Figure 17, Elastic DL can be integrated with various adversarial training methods (PGD-AT, TRADES-2.0/0.2, HAT) to consistently enhance performance.

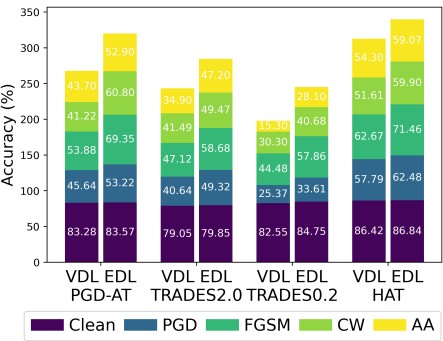

Figure 17: Different adversarial training. Our Elastic DL is orthogonal to existing adversarial training methods and can be combined with them to further improve the performance.

### D.3.3 DIFFERENT BUDGET MEASUREMENT

In addition to $\ell_\infty$-norm attack (PGD-$\ell_\infty$), we also validate the consistent effectiveness of our Elastic DL with $\ell_2$-norm (PGD-$\ell_2$) and $\ell_1$-norm (SparseFool) attacks in the Figure 18 and Table 11.

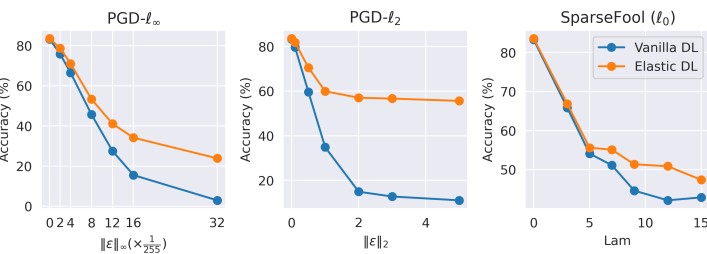

Figure 18: Different attack measurements. Our Elastic DL consistently outperforms Vanilla DL across attacks (PGD-$\ell_\infty$, PGD-$\ell_2$, SparseFool) evaluated under various metrics ($\ell_\infty, \ell_2, \ell_0$ norms).

Table 11: Adversarial robustness on CIFAR10 with different budget measurements.

| PGD $\|\cdot\|_\infty \setminus$ BUDGET | 0 | 2/255 | 4/255 | 8/255 | 12/255 | 16/255 | 32/255 | |
|---|---|---|---|---|---|---|---|---|
| VANILLA DL + RESNWT18 | 83.29 | 75.86 | 66.52 | 45.66 | 27.5 | 15.48 | 2.89 | |
| PGD-AT+ EDL - RESNET18 | 83.57 | 78.76 | 71.01 | 53.29 | 41.1 | 34.13 | 23.84 | |
| PGDL2 $\|\cdot\|_2^2 \setminus$ BUDGET | 0 | 0.1 | 0.5 | 1.0 | 2.0 | 3.0 | 5.0 | |
| VANILLA DL + RESNWT18 | 83.29 | 79.67 | 59.64 | 34.86 | 14.91 | 12.75 | 11.05 | |
| PGD-AT+ EDL - RESNET18 | 83.57 | 81.83 | 70.55 | 59.95 | 57.03 | 56.65 | 55.62 | |
| SPARSEFOOL $\|\cdot\|_0 \setminus$ LAM | 0 | 3 | 5 | 7 | 9 | 12 | 15 | 20 |
| VANILLA DL + RESNWT18 | 83.29 | 65.83 | 54.11 | 51.12 | 44.63 | 42.14 | 42.89 | 41.39 |
| PGD-AT+ EDL - RESNET18 | 83.57 | 66.83 | 55.61 | 55.11 | 51.37 | 50.87 | 47.38 | 47.13 |

### D.3.4 ZERO-ORDER GRADIENT ANALYSIS

To further validate that our method does not introduce obfuscated gradients, we use a zero-order method to estimate the gradient ($\frac{\partial f}{\partial x} \approx \frac{f(x+\epsilon)-f(x)}{\epsilon}$) and compare it with the gradient computed by autograd. The results in Table 12 show that the relative difference between the gradients computed

by autograd and the zero-order method ($\frac{|\text{Grad}_{\text{zero}} - \text{Grad}_{\text{auto}}|}{|\text{Grad}_{\text{auto}}|}$) is negligible. Moreover, the error does not accumulate or increase with the number of model layers, confirming that our method does not introduce gradient-related issues.

| Num. of Layers | 3 | 5 | 8 | 10 | 15 | 20 | 25 | 30 | 40 | 50 |
|---|---|---|---|---|---|---|---|---|---|---|
| Relative Error | 0.00066 | 0.00095 | 0.00152 | 0.00092 | 0.00118 | 0.00098 | 0.00112 | 0.00082 | 0.00060 | 0.00093 |

Table 12: Zero-order gradient analysis.

### D.3.5 HIDDEN EMBEDDING VISUALIZATION

We conduct visualization analyses on the hidden embedding to obtain better insight into the effectiveness of our proposed Elastic DL. We begin by quantifying the relative difference between clean embeddings ($x$ or $z_i$) and attacked embeddings ($x'$ or $z_i'$) across all layers. As shown in Figure 19 and Figure 20, the presence of adversarial perturbations can disrupt the hidden embedding patterns, leading to incorrect predictions in the case of Vanilla DL. In contrast, our Elastic DL appears to lessen the effects of such perturbations and maintain predicting groundtruth label.

Here are instances of CAT, SHIP, FROG, AUTOMOBILE, and TRUCK:

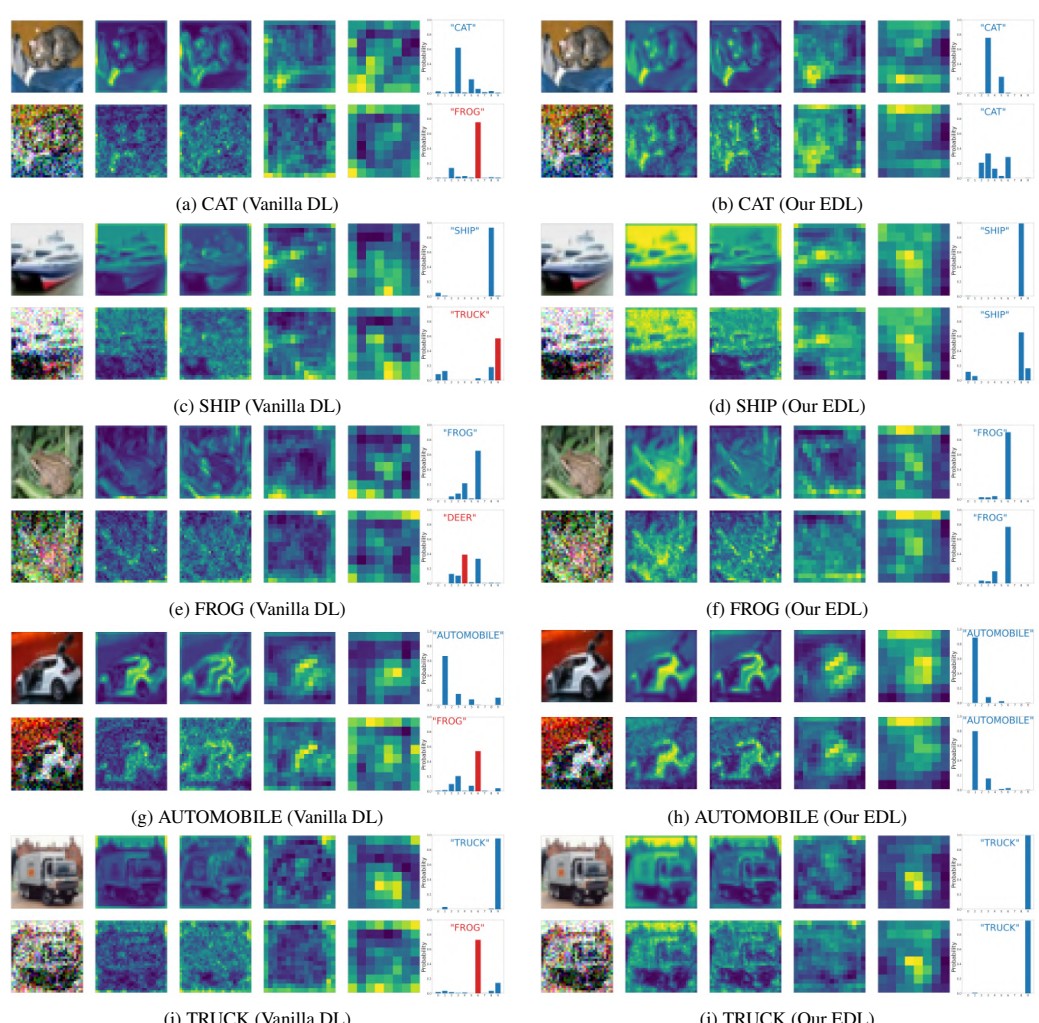

(a) CAT (Vanilla DL)    (b) CAT (Our EDL)

(c) SHIP (Vanilla DL)    (d) SHIP (Our EDL)

(e) FROG (Vanilla DL)    (f) FROG (Our EDL)

(g) AUTOMOBILE (Vanilla DL)    (h) AUTOMOBILE (Our EDL)

(i) TRUCK (Vanilla DL)    (j) TRUCK (Our EDL)

Figure 19: Hidden embedding visualization. (Part 1)

Here are instances of BIRD, HORSE, AIRPLANE, DEER and DOG:

(a) BIRD (Vanilla DL)

(b) BIRD (Our EDL)

(c) HORSE (Vanilla DL)

(d) HORSE (Our EDL)

(e) AIRPLANE (Vanilla DL)

(f) AIRPLANE (Our EDL)

(g) DEER (Vanilla DL)

(h) DEER (Our EDL)

(i) DOG (Vanilla DL)

(j) DOG (Our EDL)

Figure 20: Hidden embedding visualization. (Part 2)

### D.3.6 RECONSTRUCTION PROCESS

**Image & noise reconstruction.** In conventional feedforward neural networks, adding a perturbation $\epsilon$ to the input can lead the model to make incorrect predictions. However, as illustrated in Figure 21, our approach aims to reconstruct both the clean image $x$ and the perturbation $\epsilon$ through a dictionary learning process. To evaluate the effectiveness of our method, we quantify the reconstruction error between the recovered noise $\hat{\epsilon}$ in our Elastic DL framework and noise generated by various methods (random noise, transfer noise from ResNet/Vanilla DL, and adaptive noise from Elastic DL). As shown in Table 13, the recovered noise from our approach exhibits the smallest difference compared to the adaptive noise in Elastic DL. This result demonstrates that our proposed framework more effectively reconstructs the noise and mitigates its impact on predictions.

Table 13: Reconstruction Error. We quantify the reconstruction error between the recovered noise $\hat{\epsilon}$ and various input noises, including random noise ($\epsilon_{\text{random}}$), transfer noise from ResNet ($\epsilon_{\text{resnet}}$) and Vanilla DL ($\epsilon_{\text{vanilla}}$), as well as adaptive noise from our Elastic DL ($\epsilon_{\text{elastic}}$). Our Elastic DL demonstrates the smallest reconstruction error, indicating that our approach can adaptively recover and neutralize the input perturbation, thereby mitigating its impact.

| ERROR | $\|\cdot\|_1$ | $\|\cdot\|_2$ | $\|\cdot\|_\infty$ |
|---|---|---|---|
| $\epsilon_{\text{RANDOM}} - \hat{\epsilon}$ | $1294.75 \pm 406.78$ | $26.09 \pm 7.04$ | $0.901 \pm 0.10$ |
| $\epsilon_{\text{RESNET}} - \hat{\epsilon}$ | $131.51 \pm 10.53$ | $2.93 \pm 0.22$ | $0.163 \pm 0.01$ |
| $\epsilon_{\text{VANILLA}} - \hat{\epsilon}$ | $129.07 \pm 13.22$ | $2.85 \pm 0.26$ | $0.157 \pm 0.01$ |
| $\epsilon_{\text{ELASTIC}} - \hat{\epsilon}$ | $\mathbf{122.62 \pm 9.92}$ | $\mathbf{2.69 \pm 0.22}$ | $\mathbf{0.149 \pm 0.01}$ |

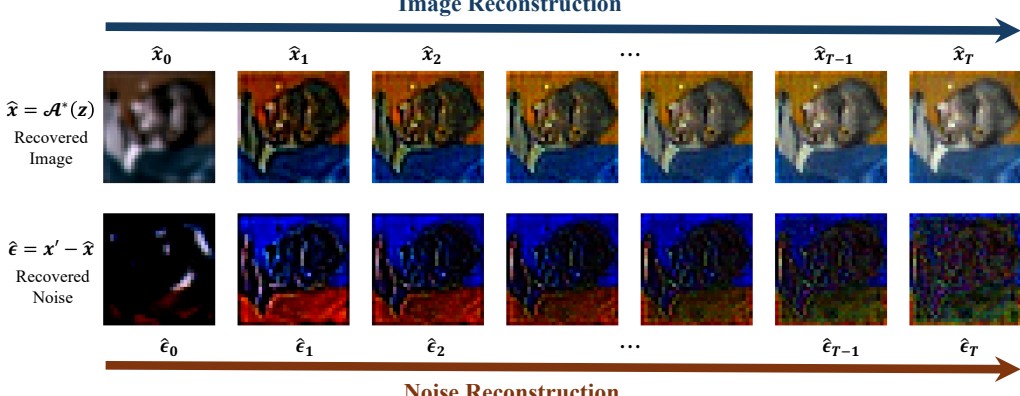

Figure 21: Reconstruction process.

Here are instances of reconstruction process in ImageNet:

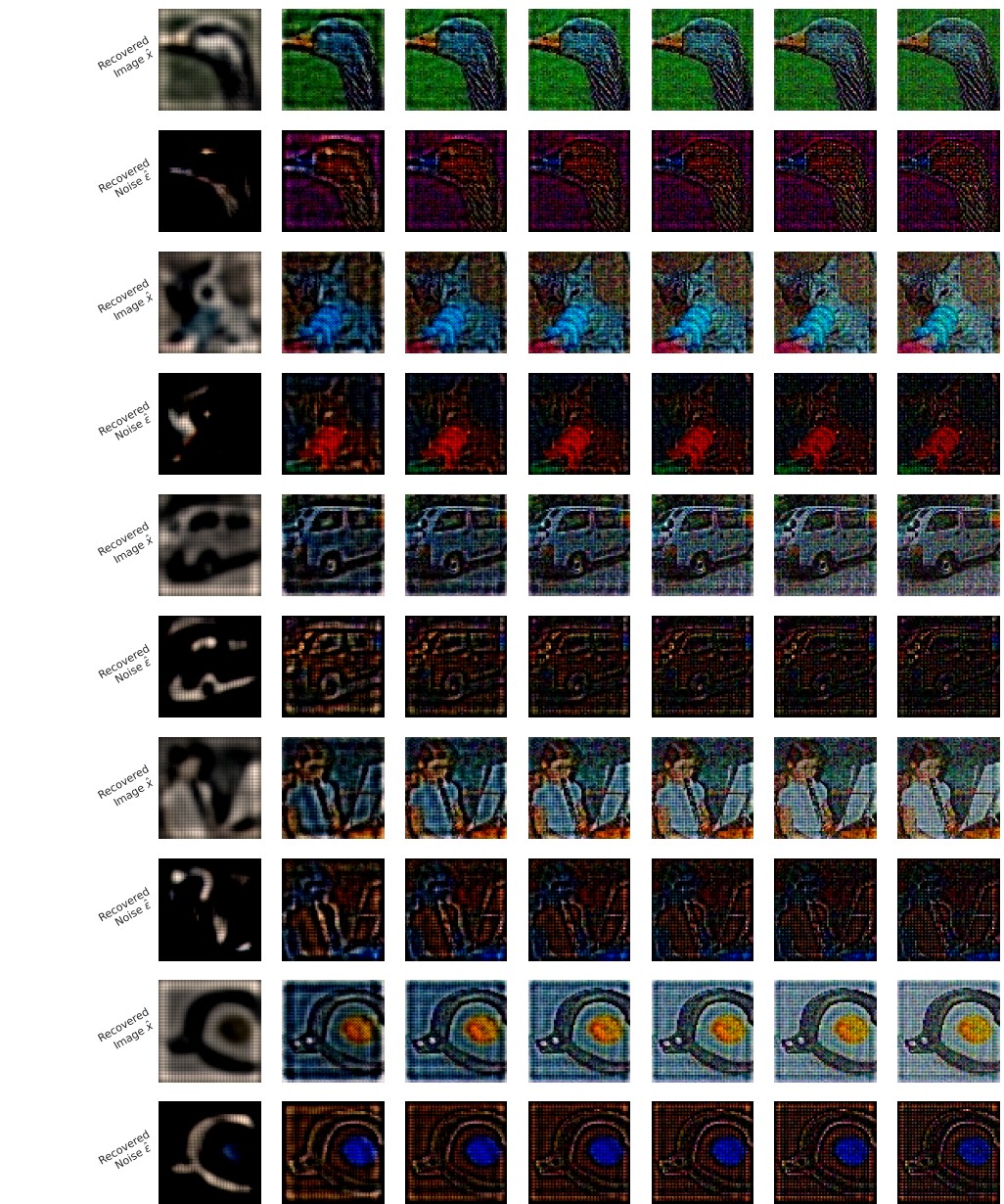

Figure 22: Reconstruction process (ImageNet)

Here are instances of reconstruction process in CIFAR10 (Part1):

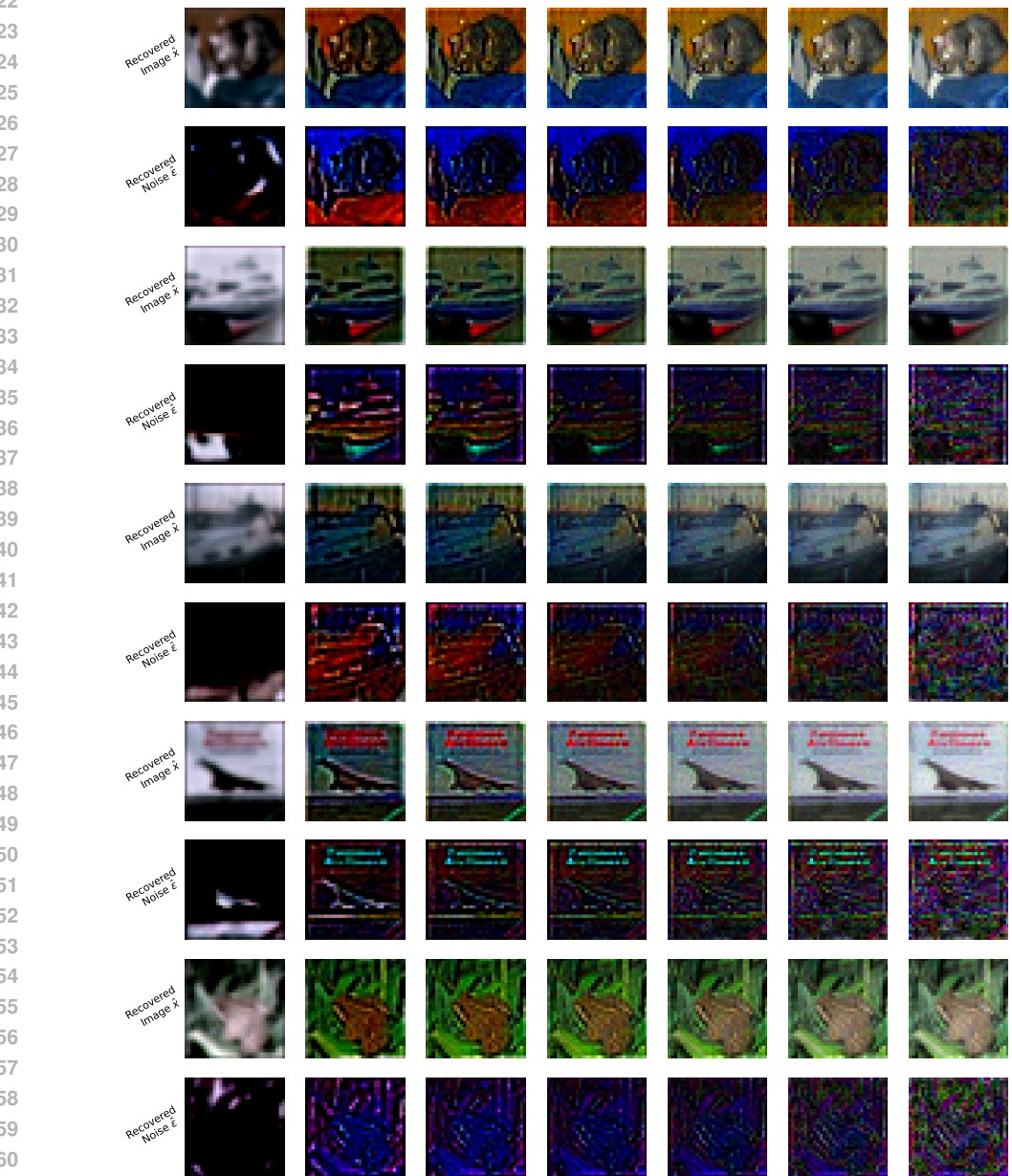

Figure 23: Reconstruction process (CIFAR10, Part 1)

Here are instances of reconstruction process in CIFAR10 (Part2):

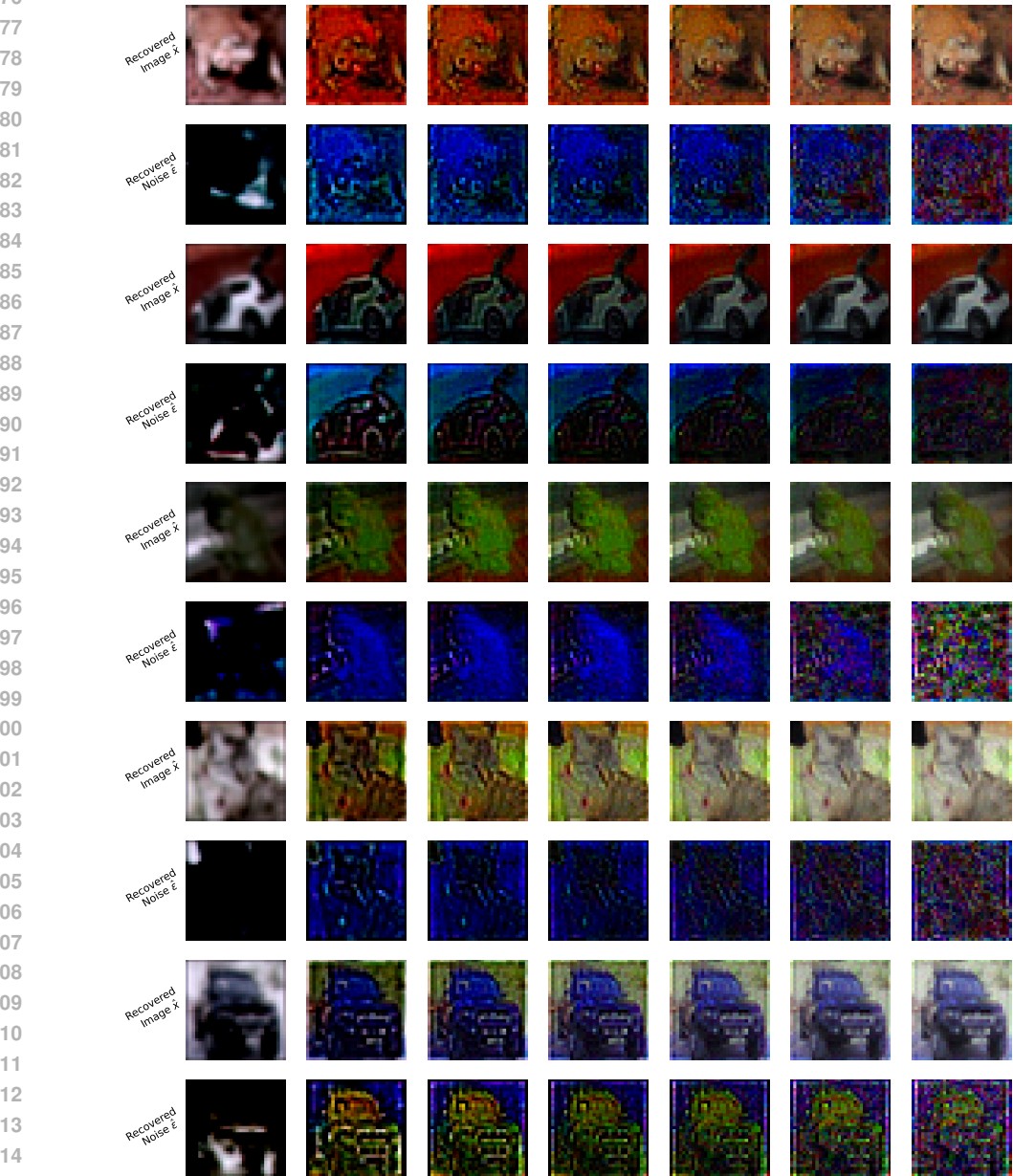

Figure 24: Reconstruction process (CIFAR10, Part 2)

