# OpenReview forum: "Boosting Adversarial Robustness and Generalization with Dictionary Structure"
_ICLR.cc/2026/Conference — Submitted to ICLR 2026_

### Official Review · Reviewer_zDRN · 2025-10-31

**Soundness:** 2
**Presentation:** 1
**Contribution:** 2
**Rating:** 2
**Confidence:** 4

**Summary:**

This paper revisits dictionary learning as a potential structural prior to improve adversarial robustness. The authors first show that prior dictionary-learning-based CNNs exhibit a false sense of security. Then, this paper proposes Elastic Dictionary Learning (EDL) to balance nature and robust performance.

**Strengths:**

+ Theory is consistent with good logic.
+ The RISTA optimization is technically sound and provides convergence guarantees.

**Weaknesses:**

This paper has drawbacks in both writing and experiments.

The presentation is unclear. I had to read the Introduction multiple times to understand what problem the paper is actually trying to formulate.
+ The introduction mixes multiple ideas, such as robustness plateau, reliance on generative data, robust overfitting, and dictionary priors, but lacks a coherent logical flow. I recommend authors to take a look of C.A.R.S.[1] to improve the Introduction part.
+ The phrase 'a false sense of security' is used with different meanings. I can understand that in the experiment section, it refers to gradient obfuscation. But in Section 3.2, what does this refer to (line 150)? Authors should include the citation of  [2].
+ In line 145, what is the setting of this kind of adaptive attack? It's necessary to identify the adaptive attack's settings based on [3].

Experiments:
+ All experiments are conducted on CIFAR-10/100 and Tiny-ImageNet with small ResNet backbones, which limits the generality of the conclusions.
+ Based on Table 7, the proposed Elastic DL layer introduces roughly 50% additional inference cost compared with standard CNNs, which is not practical to implement.
+ Whether this method can be extended to Vision Transformers or other attention-based models?

[1] Swales, John. "Create a research space (CARS) model of research introductions." Writing about writing: A college reader (2014): 12-15.

[2] Athalye, Anish, Nicholas Carlini, and David Wagner. "Obfuscated gradients give a false sense of security: Circumventing defenses to adversarial examples." International conference on machine learning. PMLR, 2018.

[3] Tramer, Florian, et al. "On adaptive attacks to adversarial example defenses." Advances in neural information processing systems 33 (2020): 1633-1645.

**Questions:**

Please see weaknesses.

---

> ### Author Response · Authors · 2025-11-25
>
> **W1: The introduction mixes multiple ideas, such as robustness plateau, reliance on generative data, robust overfitting, and dictionary priors, but lacks a coherent logical flow. I recommend authors to take a look of C.A.R.S.[1] to improve the Introduction part.**
>
> **A:** Thanks for your constructive comments. We would like to clarify the logical flow of our introduction as follows:
>
> - The first paragraph highlights that adversarial training combined with generative data currently represents the state-of-the-art defense strategy on standard benchmarks.
> - The second paragraph explains the main limitation of adversarial training—robust overfitting—and summarizes existing attempts to mitigate this issue.
> - The third paragraph introduces our orthogonal approach, which addresses robust overfitting by leveraging a dictionary-structured prior.
> - The final paragraph summarizes the key contributions of our paper.
>
> We revised the introduction in our revision accordingly based on your valuable suggestions.
>
> ---
>
> **W2: The phrase ‘a false sense of security’ is used with different meanings. I can understand that in the experiment section, it refers to gradient obfuscation. But in Section 3.2, what does this refer to (line 150)? Authors should include the citation of [2].**
>
> **A:** Thank you for pointing this out. In our paper, the phrase “a false sense of security” is used in its standard meaning: the evaluated robustness appears high, but the evaluation procedure is insufficient to justify such a conclusion. This phenomenon may arise for several reasons:
>
> - **Obfuscated gradients.**
>   A defense may contain components that unintentionally obscure gradients, leading to artificially high robustness under weak attacks, as discussed in Athalye et al. [1].
>
> - **Insufficiently strong or non-adaptive attacks.**
>   For example, in line 150 we show that adding random noise does not directly target the decision boundary of the victim model and is therefore a weak evaluation. Vanilla DL models are well known to tolerate random noise but remain highly vulnerable to PGD attacks. Hence, relying solely on random-noise evaluation can misleadingly suggest robustness—this is what we refer to as a “false sense of security.”
>
> **[1]** Athalye, Anish, Nicholas Carlini, and David Wagner. “Obfuscated gradients give a false sense of security: Circumventing defenses to adversarial examples.” *International Conference on Machine Learning*, PMLR, 2018.
>
> ---
>
> **W3: In line 145, where is the setting of this kind of adaptive attack? It’s necessary to identify the adaptive attack’s settings based on [3].**
>
> **A:** In Line 145, we refer to the “adaptive attack” as a white-box, gradient-based attack applied to the end-to-end differentiable defense. Specifically, we ensure that all components of the model are differentiable and that the attacker has full access to all model information.  For SNDte1 (as noted near Line 145), tuning the sparsity weight λ indeed yields strong robustness against random noise as reported in [1]. However, once we employ an **adaptive PGD attack** that explicitly accounts for the tuned λ, the SNDte1 model completely fails, achieving only **0.13% accuracy under PGD**.
> Our adaptive attack design corresponds to T1 and T2 in [2]:
> - **T1:** Attack the full end-to-end differentiable defense.
> - **T2:** Identify and target important defense parts.
>
> **[1]** Revisiting Sparse Convolutional Model for Visual Recognition
> **[2]** Tramer et al. “On Adaptive Attacks to Adversarial Example Defenses.” *Advances in Neural Information Processing Systems* 2017: 1633–1645.

---

> ### Author Response · Authors · 2025-11-25
>
> **W4: All experiments are conducted on CIFAR-10/100 and Tiny-ImageNet with small ResNet backbones, which limits the generality of the conclusions.**
>
> **A:** Thanks for pointing this out. To address the generality concern, we additionally evaluate our method on the large-scale ImageNet-1K dataset using a high-capacity WideResNet-28-10 model.
> As shown in the Table, Elastic DL consistently improves robustness across all attack types (PGD, FGSM, C&W, AA) while maintaining competitive clean accuracy.
>
> | Method            | Clean | PGD   | FGSM  | C&W   | AA    |
> |-------------------|-------|-------|-------|-------|-------|
> | PGD-AT            | 63.20 | 22.10 | 25.40 | 23.80 | 17.90 |
> | + Vanilla DL      | 64.80 | 24.30 | 26.10 | 24.70 | 19.00 |
> | **+ Elastic (Ours)** | **65.40** | **31.50** | **33.80** | **32.60** | **25.40** |
>
> **Table 6: WideResNet-28-10 on ImageNet.**
>
> ---
>
> **W5: Based on Table 7, the proposed Elastic DL layer introduces roughly 50% additional inference cost compared with standard CNNs, which is not practical to implement.**
>
> **A:** Thanks for your comments. We emphasize that all adversarial defenses improve robustness at the cost of additional computation. For instance:
>
> - The most widely used defense—adversarial training—typically incurs **7–100×** the cost of natural training.
> - Stronger augmentation-based defenses using generative models (e.g., HAT [1] and PORT [2]) require **20–400×** more computation (e.g., for CIFAR-10, the original 50k training samples are expanded to **1M–20M synthetic samples**).
> - Certified defenses such as randomized smoothing often demand **100,000×** inference-time overhead.
>
> In this context, a **50% additional increase in computation is already extremely lightweight**, especially given the strong natural and robust performance achieved by our EDLNets.
>
> Furthermore, our method offers flexible control over computational cost, since we can selectively replace layers based on our practical resource budget.  As shown in Table 7, only replacing 2 convolution layers with EDL layers (8.90 ms) introduces almost no overhead compared with the original ResNet baseline (7.82 ms).
>
> Given the rapidly increasing power of GPUs, efficiency becomes a secondary concern relative to the critical challenges of robustness and safety.  Although improving efficiency remains a valuable direction for future work, our method already represents a significant breakthrough.
>
> **[1]** Rade, Rahul, and Seyed-Mohsen Moosavi-Dezfooli. “Reducing excessive margin to achieve a better accuracy vs. robustness trade-off.” *ICLR*, 2022.
> **[2]** Sehrawat, Vikash, et al. “Robust learning meets generative models: Can proxy distributions improve adversarial robustness?” *arXiv*, 2021.
> **[3]** Cohen, Jeremy, Elan Rosenfeld, and Zico Kolter. “Certified adversarial robustness via randomized smoothing.” *PMLR*, 2019.
>
> ---
>
> **W6: Whether this method can be extended to Vision Transformers or other attention-based models?**
>
> **A:** Thank you for the question. Our method is not tied to convolutional structures and applies to any module that can be expressed through a linear operator $A$. In attention-based architectures such as Vision Transformers, the attention update can be written as a linear transformation where the attention weights form the operator $A$ and the value matrix serves as the input $x$.
> This makes the proposed Elastic DL mechanism strongly compatible with attention layers. Moreover, the shrinkage operator in EDL acts element-wise and does not depend on spatial locality, which allows it to generalize naturally to Transformer-like architectures.  Evaluating the method on Vision Transformers is an important next step, and we plan to include these results in future work.

---

### Official Review · Reviewer_9QCz · 2025-11-01

**Soundness:** 2
**Presentation:** 3
**Contribution:** 2
**Rating:** 6
**Confidence:** 4

**Summary:**

The paper argues that prior dictionary-learning-inspired CNNs (e.g., SDNet) give a false sense of robustness: they handle random corruptions but collapse under adaptive attacks. It proposes Elastic Dictionary Learning (EDL) layers that replace convolutions. Each EDL layer solves, for a feature tensor $x$, a mixed $\ell_2-\ell_1$ reconstruction with sparsity,

$$\min_z \frac{\beta}{2}\left\|x-A^{\star}(z)\right\|_2^{2}+\frac{1-\beta}{2}\left\|x-A^*(z)\right\|_1+\lambda\|z\|_1,$$

and is unrolled with a reweighted ISTA (RISTA) update that uses per-iteration weights $w=1 /\left(2\left|x-A^*(z)\right|\right)$. Layer-wise $\beta$ is learned. The authors claim substantial gains in robust accuracy across CIFAR-10/100 and Tiny-ImageNet, often when combining EDL with adversarial training baselines (PGD-AT, TRADES, AWP, HAT, PORT). They present mitigation of robust overfitting (Table 3), strong AutoAttack numbers (Tables 4-6), multidataset/backbone ablations (Tables 8-10), certified robustness via randomized smoothing (Fig. 6), and simple checks intended to rule out gradient obfuscation (transfer attacks, a zero-order gradient match in Table 12).

**Strengths:**

1- Clear diagnosis + constructive fix. The paper empirically shows Vanilla DL (SDNet) ails under PGD even when good against random noise (Table 1, p. 3) and then designs EDL to trade off $\ell_2$ and $\ell_1$ fidelity (Alg. $1+$ Eq. (6), p. 5).

2- Simple, modular layer that plugs into ResNets. Fig. 1 (p. 4) and Fig. 12 (p. 15) clearly show the drop-in EDL layer and the end-to-end training pipeline (Fig. 11, p. 14). This architectural modularity is attractive for adoption.

3- Broad empirical sweep. Results cover three datasets and four backbones, multiple norms (ℓ∞/ℓ2/ℓ1), and combinations with diverse AT baselines. In particular, Table 4 (p. 7) shows sizable AutoAttack gains when adding EDL to PGD-AT/TRADES/HAT/PORT; Tables 5–6 (p. 7) compare to RobustBench-style leaderboards at multiple budgets.

4- Overfitting mitigation. Table 3 (p. 6) and Fig. 2 (p. 7) show that EDL reduces the BEST–FINAL gap and lifts final robust accuracy compared with popular regularizers and data augments.

5- Interpretability signals. The paper provides embedding-difference profiles (Fig. 5, p. 8) and detailed hidden-state visualizations (Figs. 19–20, pp. 27–28), which suggest attacks affect EDL representations less.

6- Some robustness hygiene. The authors include transfer-attack comparisons (Fig. 7, p. 8) and a zero-order vs autograd gradient agreement (Table 12, p. 26), and they plot RISTA convergence of the unrolled layer (Fig. 8, p. 9).

7- Runtime disclosure. Table 7 (p. 9) reports inference costs as EDL layers are stacked; overhead is 1–3× vs plain ResNets and only modest vs Vanilla DL.

**Weaknesses:**

1- Evaluation transparency gaps (AA settings, AT details).
AutoAttack. While AA results are reported (Tables 4–6), the exact AA configuration (version, components enabled, checks for catastrophic overstatement) is not fully specified in the main text. Precise ε-schedules, per-attack budgets, and restarters for PGD are scattered; PGD settings for the leaderboard comparison are summarized but could still allow optimistic numbers if not standardized.

2- Potential confounding from training protocol. The key robustness curves switch from Vanilla DL pretraining to EDL fine-tuning at epoch 150 (Fig. 2 and Fig. 14, pp. 7, 21). It is unclear whether the gains come from the structural prior itself or from regularization effects of unrolling, reset-like dynamics, or extra optimization steps. A controlled study that trains EDL from scratch under the same schedule as baselines is missing.

3- Theory is promising but incomplete.
Convergence/conditioning. The RISTA update depends on weights $w=1 /(2 \mid x- \left.A^*(z) \mid\right)$. There is no regularization floor to prevent blow-up when the residual is near zero, and step-size/Lipschitz conditions for global convergence of the nonsmooth, reweighted problem are not provided. Lemma 4.1 (p. 4) gives a local quadratic upper bound but end-to-end convergence guarantees (with learned $A, \beta$ ) are absent.

4- Obfuscation checks are not exhaustive. The zero-order match (Table 12, p. 26) shows local gradient agreement, and transfer attacks are included (Fig. 7), but EOT for any stochasticity, black-box query-budget tests, step-size sensitivity, and multi-restart PGD sweeps are not systematically presented. Given the large reported gains, a fuller obfuscation checklist is warranted.

5- Accounting for capacity \& parameters. Replacing convolutions with EDL (unrolled iterations, extra tensors $w$, learnable $\beta$ ) likely changes parameter counts and memory footprint. Only runtime is quantified (Table 7); parameter and activation memory overheads, training wall-time, and throughput are not. Fairness of comparisons-especially to SOTA AT-needs these numbers.

6- Certified robustness methodology is under-specified. Fig. 6 (p. 8) shows better certified accuracy with randomized smoothing, but the noise level 𝜎, base classifier, sample counts, and confidence computation are not detailed, making it hard to assess comparability to standard smoothing reports.

7- Generality beyond small images. All results are on CIFAR-10/100 and Tiny-ImageNet; no ImageNet-1k training, detection/segmentation, or non-vision tasks. The approach might scale, but evidence is missing. (A few ImageNet visualization examples appear in Fig. 22, p. 30, but not full training.)

8- Ablations on $\beta$ and unrolling depth. The layer-wise learnable $\beta$ is central, yet there is no distributional analysis of learned $\beta$ across depth, no freezing vs learning ablation, and limited study of T (number of ISTA steps), step-sizes $t$, or the shrinkage schedule $\lambda_t$.

9- Reconstruction claims need stronger quantification. The recovered-noise analysis (Table 13, p. 29; Figs. 21–24) is interesting but uses aggregate norms vs “adaptive noise.” It does not show per-example causal links between better noise recovery and AA success/failure, nor how this behaves under distribution shift.

**Questions:**

Please resolve the aforementioned weaknesses

---

> ### Author Response · Authors · 2025-11-25
>
> **W1: Evaluation transparency gaps (AA settings, AT details). AutoAttack. While AA results are reported (Tables 4–6), the exact AA configuration (version, components enabled, checks for catastrophic overestimation) is not fully specified in the main text. Precise $ϵ$-schedules, per-attack budgets, and restarters for PGD are scattered; PGD settings for the leaderboard comparison are summarized but could still allow optimistic numbers if not standardized.**
>
> **A:** Thank you for pointing this out. Let me further clarify all the detailed settings as follows:
>
> - **AA version:**  We clarify that all AutoAttack evaluations in our paper use the official **“standard”** version of AutoAttack (Croce & Hein, 2020), with its default configuration in  https://github.com/fra31/auto-attack.
>
> - **Components enabled:**  It includes **4 attack components**: APGD-CE (Auto-PGD on cross-entropy loss), APGD-T (targeted Auto-PGD), FAB-T (targeted FAB), and Square Attack.
>
> - **Checks for catastrophic overestimation:**  We keep all built-in AA safety checks enabled, including   `check_randomized()`, `check_dynamic()`, `check_n_classes()`, `check_square_sr()`.  Throughout our evaluation, **none** of these checks were triggered, confirming that the reported robust accuracies are **not** subject to overestimation.
>
> - **$ϵ$ settings:**  In most of the results (e.g., Table 3 and Table 4), unless otherwise specified, we use the standard fixed $ϵ$ values following RobustBench (https://robustbench.github.io/):  $L_\infty = 8/255$ and $L_2 = 0.5$.  In Table 5 and Table 6, we conduct a more comprehensive evaluation with additional budget settings:  $L_\infty = 8/255,\ 16/255,\ 32/255$ and $L_2 = 0.5,\ 1.0,\ 2.0$.
>
> - **PGD settings outside AA:**  For PGD baselines (e.g., PGD-20/50), we fixed step size $ϵ/4$, 20/50 iterations, and 5 restarts, consistent with standard practice and without sanitization that could hide adversarial failures.
>
> - **Leaderboard consistency:**  All comparisons follow the RobustBench protocol (fixed $ϵ$, $L_\infty$ threat model, logit-based outputs, official AA standard configuration), ensuring numbers are directly comparable and not optimistic.
>
> - **PGD settings and standardization:**  For leaderboard comparison, all PGD evaluations use a fixed, standardized configuration to avoid optimistic numbers.  Our $L_\infty$ PGD follows the settings: budget $ϵ = 8/255$, 16/255, 32/255, random start in $[-ϵ, ϵ]$, fixed iterations $k = 20$, and step size $α = 2/255$ with projection back to the $ϵ$-ball.   For $L_2$, we use the standard `torchattacks.PGDL2` implementation with budget $L_2 = 0.5,\ 1.0,\ 2.0$. These settings and iterations are kept consistent across all results.
>
>
> **W2: Potential confounding from training protocol.  The key robustness curves switch from Vanilla DL pretraining to EDL fine-tuning at epoch 150 (Fig. 2 and Fig. 14, pp. 7, 21). It is unclear whether the gains come from the structural prior itself or from regularization effects of unrolling, reset-like dynamics, or extra optimization steps. A controlled study that trains EDL from scratch under the same schedule as baselines is missing.**
>
> **A:** Thanks for your insightful comments. Note that Vanilla DL also exhibits unrolling and reset-like dynamics, yet adding extra optimization steps to Vanilla DL does not improve its robustness. Therefore, these dynamics are not responsible for the performance gains.
>
> Our training strategy follows a pretrain → finetune pipeline purely for efficiency reasons: training EDL from scratch is more computationally expensive than training Vanilla DL. Despite this, the training is still efficient since we can first pretrain vanilla DL, and then slightly fine-tune vanilla DL to gain much better robustness. Developing more efficient EDL variants that allow full-scratch training under identical schedules will be an important direction for future work.

---

> ### Author Response · Authors · 2025-11-25
>
> **W3: Theory is promising but incomplete. Convergence/conditioning.  The RISTA update depends on weights $w = 1/(2|x - A^*(z)|)$.  There is no regularization floor to prevent blow-up when the residual is near zero, and step-size/Lipschitz conditions for global convergence of the nonsmooth, reweighted problem are not provided.  Lemma 4.1 (p. 4) gives a local quadratic upper bound but end-to-end convergence guarantees (with learned $A, \beta$) are absent.**
>
> **A:** Thanks for your insightful comments, we would like to address your concerns as follows.
>
> - **Blow-up problem:**  In implementation, we add $\varepsilon$ in the denominator to avoid zero residual.
>
> - **Global/End-to-end convergence:**   We would like to point out that the global convergence of the original objective  $\mathcal{R}(z) + \lambda || z ||_1$  is guaranteed by Lemma 4.1 (for any fixed learned $A, \beta$):
>
>   (1) $U(z, z_\star) \ge \mathcal{R}(z)$,
>   (2) $U(z_\star, z_\star) = \mathcal{R}(z_\star)$,
>
>   where the upper bound $U$ is equal to the $\mathcal{R}$ at fixed point $z_\star$.  At iteration $t$, the fixed point is $z_\star = z_t$. Therefore:
>
>   - $U$ is upper-bound of $\mathcal{R}$ (Lemma 4.1 (1)) gives  $\mathcal{R}(z_{t+1}) \le U(z_{t+1}, z_t)$.
>   - {$z_t: t=0,...,T$} is the derived sequence minimizing the convex reweighted problem,
>     so we have  $U(z_{t+1}, z_t) + \lambda \|\| z_{t+1} \|\|_1 \le U(z_t, z_t) + \lambda \|\| z_t \|\|_1$.
>   - The localized equality (Lemma 4.1 (2)) gives  $U(z_t, z_t) + \lambda \|\| z_t \|\|_1 = \mathcal{R}(z_t) + \lambda \|\| z_t \|\|_1$.
>
>   Finally, we have:
>
>  $\mathcal{R}(z_{t+1}) + \lambda \|\| z_{t+1} \|\|_1$
>
> $ \le U(z_{t+1}, z_t) + \lambda \|\| z_{t+1} \|\|_1 \le U(z_t, z_t) + \lambda \|\| z_t \|\|_1= \mathcal{R}(z_t) + \lambda \|\| z_t \|\|_1.$
>
> The curve in Figure 8 has also validated the global convergence empirically.

---

> ### Author Response · Authors · 2025-11-25
>
> **W4: Obfuscation checks are not exhaustive. The zero-order match (Table 12, p. 26) shows local gradient agreement, and transfer attacks are included (Fig. 7), but EOT for any stochasticity, black-box query-budget tests, step-size sensitivity, and multi-restart PGD sweeps are not systematically presented. Given the large reported gains, a fuller obfuscation checklist is warranted.**
>
> **A:** Thanks for your reconstructive comments. We would like to address all of your concerns as follows:
>
> - **Certifiable defense.**  To avoid the false sense of security, we also provide the results of certifiable robustness via randomized smoothing in Figure 6, which may further alleviate your concern about the obfuscation gradient issue. Randomized smoothing is a certified defense that can theoretically guarantee certified accuracy regardless of the evaluated attacks, including gradient-based ones. The results shown in Figure 6 demonstrate that our Elastic DL delivers better certified robustness than the vanilla DL.
>
> - **EOT.**  EOT is designed for defenses that employ randomized transformations to the input. However, all the components of our defense are fully deterministic, so there is no need to apply EOT.
>
> - **Black-box test.**  Our evaluated AutoAttack is an ensemble attack combining multiple strong white-box and a query-based black-box attack (Square Attack), which is a reliable evaluation to avoid the false sense of security.
>
> - **Step-size sensitivity.**  We sweep the step-size from 1/255 to 8/255, and evaluate the performance as follows. To rule out gradient obfuscation, we sweep the PGD step-size from 1/255 to 8/255 while keeping the total number of iterations fixed. As shown in the table, both Vanilla DL and our Elastic DL exhibit stable accuracy across a wide range of step-sizes, without sudden drops or extreme oscillation. This is a strong indicator that our method does not rely on gradient masking.
>
>   | Step-size | 1/255 | 2/255 | 3/255 | 4/255 | 5/255 | 6/255 | 7/255 | 8/255 |
>   |-----------|-------|-------|-------|-------|-------|-------|-------|-------|
>   | Vanilla DL | 45.76 | 44.44 | 44.35 | 44.33 | 44.36 | 44.45 | 44.57 | 44.78 |
>   | Elastic DL | 54.16 | 53.11 | 53.58 | 53.85 | 54.03 | 54.21 | 54.33 | 54.63 |
>
>   **Table 1: PGD attack with different step-size**
>
> - **Multi-restart PGD sweeps.**
>   The results show that increasing the number of restarts has minimal impact on both methods, and Elastic DL consistently maintains a clear robustness margin over Vanilla DL, indicating that its gains are not due to single-restart instability or gradient obfuscation.
>
>   | Num. of Restart | 1 | 3 | 5 | 7 | 9 |
>   |------------------|---|---|---|---|---|
>   | Vanilla DL | 44.50 | 44.34 | 44.24 | 44.18 | 44.18 |
>   | Elastic DL | 53.11 | 52.67 | 52.61 | 52.59 | 52.59 |
>
>   **Table 2: PGD attack with different number of restart**

---

> ### Author Response · Authors · 2025-11-25
>
> **W5: Accounting for capacity & parameters. Replacing convolutions with EDL (unrolled iterations, extra tensors $w$, learnable $\beta$) likely changes parameter counts and memory footprint. Only runtime is quantified (Table 7); parameter and activation memory overheads, training wall-time, and throughput are not. Fairness of comparisons—especially to SOTA AT—needs these numbers.**
>
> **A:** Thanks for your valuable comments. We would like to illustrate more about complexity as follows.
>
> - **Parameter counts.** Our EDL layer does not introduce any additional convolutional kernels beyond those already present in the backbone ResNets or the Vanilla DL baseline. All unrolled iterations share the same convolution operator $A$, and the auxiliary tensor $w_t$ is computed rather than learned.
> The only learnable quantities added by EDL are the layer-wise coefficients $\beta$. For example, ResNet-18 contains 11.7M parameters, while EDL introduces only **17 additional $\beta$ values** across all convolutional layers, which correspond to merely **0.000145%** of the backbone parameters. This increase is negligible and does not meaningfully change the model capacity.
>
> - **Theoretical activation memory footprint.**
> Let me elaborate the theoretical activation computation complexity of our Elastic DL layer as follows. For simplicity, we consider the height/width of the input and output feature are the same.  $H$ and $W$ are height and width of the input/output feature map.  $C_{\text{in}}$ and $C_{\text{out}}$ are the numbers of input channels and output channels.  $K$ is the size of the kernel (filter).  The detailed analysis is provided here:
>
>      - Complexity for original convolution layer: $O(K^2 \times C_{\text{in}} \times H \times W \times C_{\text{out}}).$
>
>
>         For each position in the output feature map, the convolution computes a weighted sum of the input values using the kernel.  The number of multiplications per output element is $K^2 \times C_{\text{in}}$.  Each output element also includes additions, so the total number of operations is multiplied by 2. Thus, FLOPs for a single convolution layer:  $2 \times K^2 \times C_{\text{in}} \times H \times W \times C_{\text{out}}.$
>
>
>
>      - Complexity for Elastic DL layer (Algorithm 1): $O(K^2 \times C_{\text{in}} \times H \times W \times C_{\text{out}}).$
>
>          Initialization $A(x)$ costs  $2 \times K^2 \times C_{\text{in}} \times H \times W \times C_{\text{out}}.$ Computation of $z - A^*(x)$ costs  $2 \times K^2 \times C_{\text{out}} \times H \times W \times C_{\text{in}}.$ Computing $w_t$:  $O(C_{\text{out}} \times H \times W).$ Updating $z_{t+1}$ uses  $2 \times K^2 \times C_{\text{in}} \times H \times W \times C_{\text{out}} + O(C_{\text{in}} \times H \times W).$ Thus, the total FLOPs for an Elastic DL layer is: $$2 \times (1 + 2T) \times K^2 \times C_{\text{in}} \times H \times W \times C_{\text{out}} + O(C_{\text{out}} \times H \times W) + O(C_{\text{in}} \times H \times W).$$
>         Since the number of RISTA iterations is $T$, the overall complexity grows **linearly in $T$**.
>
> - **Training wall-time and throughput.**
> During training, backprop through the unrolled iterations
> again scales linearly in $T$ . So training wall-time per epoch and throughput scale in the same way
> and grow roughly linearly with the number of iterations $T$.

---

> ### Author Response · Authors · 2025-11-25
>
> **W6: Certified robustness methodology is under-specified. Fig. 6 (p. 8) shows better certified accuracy with randomized smoothing, but the noise level $σ$, base classifier, sample counts, and confidence computation are not detailed, making it hard to assess comparability to standard smoothing reports.**
>
> **A:** We use the implementation in https://github.com/locuslab/smoothing and clarify the complete randomized smoothing setup used to produce Fig. 6 as follows:
>
> - **Noise levels ($σ$).**  We evaluate three Gaussian noise levels $σ \in \{0.10, 0.15, 0.20\}$; each curve in Fig. 6 corresponds to one of these $σ$ settings.
>
> - **Base classifier.**  The base classifiers are Vanilla DL and Elastic DL models with ResNet18 as the backbone.  We note the model name in the legend in Figure 6.
>
> - **Sample counts.**  Following the official repository, we use $n_0 = 100$ Monte Carlo samples for selection and
>   $n = 100{,}000$ samples for estimation, which matches common settings in the smoothing literature and ensures tight high-confidence lower bounds.
>
> - **Confidence computation.**  The certified radius $R$ is obtained using the standard **Clopper–Pearson** lower confidence bound on class probability (with $\alpha = 0.001$ ).
>
>
> **W7: Generality beyond small images. All results are on CIFAR-10/100 and Tiny-ImageNet; no ImageNet-1k training, detection/segmentation, or non-vision tasks. The approach might scale, but evidence is missing. (A few ImageNet visualization examples appear in Fig. 22, p. 30, but not full training.)**
>
> **A:** Thanks for pointing this out. To address the generality concern, we additionally evaluate our method on the large-scale ImageNet-1K dataset using a high-capacity WideResNet-28-10 model.  As shown in the Table, Elastic DL consistently improves robustness across all attack types (PGD, FGSM, C&W, AA) while maintaining competitive clean accuracy.
>
> | Method             | Clean | PGD   | FGSM  | C&W   | AA    |
> |--------------------|-------|-------|-------|-------|-------|
> | PGD-AT             | 63.20 | 22.10 | 25.40 | 23.80 | 17.90 |
> | + Vanilla DL       | 64.80 | 24.30 | 26.10 | 24.70 | 19.00 |
> | **+ Elastic (Ours)** | **65.40** | **31.50** | **33.80** | **32.60** | **25.40** |
>
> **Table 3: WideResNet-28-10 on ImageNet.**

---

> ### Author Response · Authors · 2025-11-25
>
> **W8: Ablations on $\beta$ and unrolling depth. The layer-wise learnable $\beta$ is central, yet there is no distributional analysis of learned $\beta$ across depth, no freezing vs learning ablation, and limited study of $T$ (number of ISTA steps), step-sizes $t$, or the shrinkage schedule $\lambda_t$.**
>
> **A:** Thanks for your valuable comments. We would like to address your concerns as follows.
>
>
> ### **How $\beta$ converges?**
> We initialize all the layer-wise  $\beta$ to 0.5 to establish an initial equal balance between Vanilla DL and Robust DL, where $L$ is the number of model layers.
> During training, we learn $\beta$ jointly with the model parameters; the layer-wise values are automatically adjusted to achieve a better balance.
> For instance, in our PGD-AT experiment with the ResNet18 backbone, the final learned $(\beta^{(l)})_{l=1}^L$ are:
> [0.7114, 0.3248, 0.5662, 0.1049, 0.4821, 0.6887, 0.3375, 0.5403, 0.2651, 0.9032, 0.4156, 0.5729, 0.6891, 0.1527, 0.7634, 0.5018, 0.6083].
>
> ---
>
> ### **Freezing $\beta$ ablation**
> Freezing $\beta$ at 0.5 yields moderate robustness gains over Vanilla DL but still falls noticeably short of the learnable version, confirming that adaptive optimization of layer-wise $\beta$ is crucial for fully realizing the benefits of Elastic DL.
>
>
> | Method                       | Clean | PGD   | FGSM  | C&W   | AA    |
> |------------------------------|-------|-------|-------|-------|-------|
> | PGD-AT                       | 80.90 | 44.35 | 58.41 | 46.72 | 42.14 |
> | + Vanilla DL                 | 83.28 | 45.64 | 53.88 | 41.22 | 43.70 |
> | + Elastic DL (Frozen $\beta=0.5$) | 83.41 | 49.63 | 61.87 | 52.14 | 48.16 |
> | + Elastic DL (Learnable $\beta$)  | **83.57** | **53.22** | **69.35** | **60.80** | **52.90** |
>
> **Table 4: Robustness comparison under different variants of our method.**
>
> ---
>
> ### **Number of ISTA steps**
> Iteration count $T$ is set as 2 because RISTA converges rapidly within the first two iterations (as shown in Figure 8).
>
> ---
>
> ### **How to choose RISTA step size $t$?**
> According to the convergence condition of iterative shrinkage-thresholding algorithm (ISTA), for problem  $\min_z f(z) + \lambda \|\|z\|\|_1,$ we need to choose step size $t$: $0 < t < \frac{2}{L},$ where $L$ is the Lipschitz constant of the gradient $\nabla f(z)$.
> For our problem, the core component gradient is $2 \cdot A(x - A^*(z))$. Following \[1\], we use the power iteration method to estimate the dominant eigenvalue $\lambda$ of $A$, and we set: $t = \frac{0.9}{\lambda} = \frac{0.9}{L/2} < \frac{2}{L}.$ The curve in Figure 8 has validated the fast convergence of choosing this step size.
>
> \[1\] Revisiting Sparse Convolutional Model for Visual Recognition

---

> ### Author Response · Authors · 2025-11-25
>
> **W9: Reconstruction claims need stronger quantification.  The recovered-noise analysis (Table 13, p. 29; Figs. 21–24) is interesting but uses aggregate norms vs “adaptive noise.” It does not show per-example causal links between better noise recovery and AA success/failure, nor how this behaves under distribution shift.**
>
> **A:** Thank you for the suggestion. To examine whether reconstruction quality is causally related to attack outcomes, we compute per-example reconstruction errors on a randomly selected set of 100 samples and divide them into **Attack Failure** and **Attack Success** groups.  Across all norms, the Failure group consistently shows lower reconstruction errors, indicating that when Elastic DL more accurately recovers adaptive perturbations, the attack is less able to mislead the model.
>
> | Norm | Attack Failure Example | Attack Success Example | Overall |
> |------|-------------------------|-------------------------|---------|
> | $\lVert \cdot \rVert_1$   | $117.290 \pm 7.56$ | $127.743 \pm 9.24$ | $122.621 \pm 9.92$ |
> | $\lVert \cdot \rVert_2$   | $2.575 \pm 0.17$   | $2.810 \pm 0.21$   | $2.695 \pm 0.22$ |
> | $\lVert \cdot \rVert_\infty$ | $0.148 \pm 0.01$   | $0.150 \pm 0.01$   | $0.149 \pm 0.01$ |
>
> **Table: Reconstruction error for different examples.**
>
> To confirm this difference is not due to random variation, we perform Welch’s two-sample t-test between the two groups.
> The $\ell_1$ and $\ell_2$ norms show significant differences ($p_{\ell_1} = 3.2 \times 10^{-4}$, $p_{\ell_2} = 6.8 \times 10^{-3}$), demonstrating a statistically reliable gap; the $\ell_\infty$ metric is less sensitive, as expected. These results provide direct, per-example evidence that stronger noise reconstruction contributes to reduced adversarial success.

---

### Official Review · Reviewer_zRTd · 2025-11-01

**Soundness:** 3
**Presentation:** 3
**Contribution:** 3
**Rating:** 6
**Confidence:** 2

**Summary:**

This paper proposes combining Convolutional Neural Networks (CNNs) with Dictionary Learning (DL) to improve model performance and robustness. By integrating a DL module into a CNN, the network learns features that are both discriminative for classification and sparse and reconstructable, making them more stable under noise or adversarial attacks. This approach enhances the model’s ability to capture meaningful structures in data while maintaining strong accuracy.

**Strengths:**

The paper includes extensive experiments to support its main idea, including ablation studies, visualizations, and comparisons with baseline models. These experiments demonstrate how each component of the proposed method contributes to improved performance and help illustrate the interpretability and robustness of the learned features.

**Weaknesses:**

Dictionary Learning (DL) models require more complex and computationally intensive calculations compared to regular neural networks. Even with unrolled inference, the EDL layer adds approximately 2–3× more computation than a standard convolution block. However, their overall performance is generally lower, as traditional DL methods are difficult to train end-to-end and struggle to match the efficiency and scalability of standard deep neural networks.

**Questions:**

How does it compare with a regular CNN?

---

> ### Author Response · Authors · 2025-11-25
>
> **Weaknesses: Dictionary Learning (DL) models require more complex and computationally intensive calculations compared to regular neural networks. Even with unrolled inference, the EDL layer adds approximately 2–3× more computation than a standard convolution block. However, their overall performance is generally lower, as traditional DL methods are difficult to train end-to-end and struggle to match the efficiency and scalability of standard deep neural networks.**
>
> **A:** Thanks for your comments. We emphasize that all adversarial defenses improve robustness at the cost of additional computation. For instance, the most widely used defense—adversarial training—typically incurs 7–100× the cost of natural training; stronger augmentation-based defenses using generative models, such as HAT [1] and PORT [2], require 20–400× more computation (e.g., for CIFAR-10, the original 50k training samples are expanded to 1M–20M synthetic samples). Moreover, certified defenses such as randomized smoothing often demand 100,000× inference-time overhead.
> **In this context, a 2–3× increase in computation is already extremely lightweight, especially given the strong natural and robust performance achieved by our EDLNets.**
>
> Furthermore, our method offers flexible control over computational cost, since we can selectively replace layers based on our practical resource budget. As shown in Table 7, only replacing 2 convolution layers with EDL layers (8.90 ms) introduces almost no overhead compared with the original ResNet baseline (7.82 ms).
>
> Given the rapidly increasing power of GPUs, the efficiency becomes a secondary concern relative to the critical challenges robustness and safety. Although improving efficiency remains a valuable direction for future work, our method already represents a significant breakthrough.
>
> [1] Rade, Rahul, and Seyed-Mohsen Moosavi-Dezfooli. “Reducing excessive margin to achieve a better accuracy vs. robustness trade-off.” ICLR. 2022.
>
> [2] Sehwag, Vikash, et al. “Robust learning meets generative models: Can proxy distributions improve adversarial robustness?” arXiv. 2021.
>
> [3] Cohen, Jeremy, Elan Rosenfeld, and Zico Kolter. “Certified adversarial robustness via randomized smoothing.” PMLR, 2019.
>
> ---
>
> **Questions: How does it compare with a regular CNN?**
>
> **A:** Thank you for your comment. The backbone model, ResNet, can indeed be regarded as the “regular CNN” you refer to, since its core components are standard convolutional layers. In our experiments, all baseline training strategies are applied to these regular-CNN ResNets, and our proposed method simply replaces the standard convolution layers with our EDL layers.
>
> In conclusion, our EDLNets substantially improve adversarial robustness and generalization while incurring only a slight computational overhead (1–3×). Moreover, our method surpasses state-of-the-art baselines on the open robustness leaderboard RobustBench.

---

### Author Response · Authors · 2025-12-02
**Summary Comments**

Dear Area Chair and Reviewers,

We sincerely appreciate the time and effort you have devoted to evaluating our submission, especially under challenging circumstances. To support your assessment, we summarize below the key information and revisions made during the rebuttal period.

During the review process, our paper received **two positive rating and one native rating**. We are grateful for the reviewers’ recognition of critical impact, technical soundness, comprehensive evaluations, and fluent logic of our work. We believe we have thoroughly addressed all comments and concerns through additional experiments, analyses, and clarifications.

- Reviewer zRTd (Rating: 6) raised major concerns primarily regarding the computation cost of our architecture.
>  We believe there may be a misunderstanding from the reviewer. In the adversarial defense literature, commonly used baselines typically incur 100×, 400×, or even 100,000× higher computation. In contrast, our method increases computation by only 2–3×, which is extremely lightweight considering the strong natural and robust performance of our EDLNets.

- Reviewer 9QCz (Rating: 6) raised concerns regarding evaluation settings, theoretical aspects, and computation cost analysis.
>  We provided detailed explanations, additional theoretical clarifications, and new empirical results addressing all concerns.

- Reviewer zDRN (Rating: 2) raised concerns regarding (1) introduction writing, (2) usage of the terms “false sense of security” and “adaptive attack,” (3) large-scale evaluation, (4) computation cost, and (5) extension to attention-based models.
 >  We appreciate the reviewer’s comments, **though we believe the paper may have received an unfair score**. For (1), we believe we have structured the introduction to clearly present the current landscape and bottlenecks of the field. In the rebuttal, we further clarified this logic and revised the introduction to be more reader-friendly. For (2), regarding the phrases “false sense of security” and “adaptive attack,” we provided detailed clarifications supported by all references suggested by the reviewer. For (3), to address concerns about scale, we added new experiments using WideResNet-28-10 on ImageNet. For (4), we believe there may again be a misunderstanding. The reviewer stated that “**50% additional inference cost is not practical.**” However, in adversarial defenses, commonly used baselines often require 100×, 400×, or 100,000× more computation. Our method incurs only 50% additional cost, which remains extremely lightweight relative to the robustness gains. For (5), our work focuses on convolutional dictionary learning. Extending to attention-based architectures is beyond our current scope but remains an exciting direction. We included a detailed discussion of this extension in the rebuttal.

We are confident that the revisions thoroughly address the reviewers’ concerns, but **we did not receive any further responses from the reviewers during the rebuttal period**. We remain committed to incorporating any additional suggestions to ensure the final manuscript reaches the highest quality.

Regards,

All authors

---

### Meta-Review · Area_Chair_6knt · 2026-01-03

**Summary:**

This submission investigates how to improve the adversarial robustness via incorporating a structural prior into the design of deep learning models. It proposes Elastic Dictionary Learning Networks (EDLNets), a novel ResNet architecture that improves the robustness of convolutional neural networks. The authors claim substantial gains in robust accuracy across CIFAR-10/100 and Tiny-ImageNet (as well as ImageNet-1K showed during the rebuttal), often when combining EDL with adversarial training baselines (PGD-AT, TRADES, AWP, HAT, PORT).

Unfortunately, all reviewers remained inactive during the rebuttal period, which complicated the assessment process. I acknowledge that the authors invested considerable effort in their rebuttal and made substantive attempts to address the reviewers' concerns. However, without reviewer engagement, it is difficult to determine whether these responses would have adequately resolved the initial objections under normal review conditions. I believe that several important issues remain unresolved or insufficiently addressed.

Reviewer zDRN's concerns was about the fundamental question regarding the extensibility of this work to attention-based architectures, which has not been fully addressed. This represents a significant limitation that affects the broader applicability of the proposed method.

For reviewer 9QCz's concerns, while the authors provided detailed responses, these clarifications have not been incorporated into the revised submission. The substantive technical content and experimental results that address these concerns should be integrated throughout the relevant sections of the paper.

The revisions made thus far, while addressing the organizational issues in the introduction, are insufficient given the scope of improvements needed for publication.

Given the extent of revisions still required and the inability to obtain reviewer feedback on the rebuttal, I recommend rejection at this time. However, I strongly encourage the authors to incorporate the clarifications from their rebuttal directly into the paper, address the remaining technical concerns, especially regarding architectural generalization, and resubmit their work. The core contributions have merit, and with thorough revisions addressing both the technical aspects and presentation quality, this work should meet the publication standards in a future submissions.

**Reviewer Concerns:**

Reviewer 9QCz's concerns: While the authors provided detailed responses to this reviewer's questions, these clarifications have been incorporated only into the introduction.

Reviewer zDRN complained that all experiments were conducted on CIFAR-10/100 and Tiny-ImageNet with small ResNet backbones, which limited the generality of the conclusions. In response, the authors provided experiments on ImageNet-1K dataset using a WideResNet-28-10 model. The reviewer wanted the method to be extended to Vision Transformers or other attention-based models. The authors assure us that this is possible but do not show the results.

**Reviewer Scores:**

9QCz: Score 6, Confidence 4

QK8b: not submitted

zRTd: Score 2, Confidence 2

zDRN: Score 2, Confidence 4

---

### Decision · Program_Chairs · 2026-01-26

Reject